# DAGE: DAG Query Answering via Relational Combinator with Logical Constraints

## Abstract

Predicting answers to queries over knowledge graphs is called a complex reasoning task because answering a query requires subdividing it into subqueries. Existing query embedding methods use this decomposition to compute the embedding of a query as the combination of the embedding of the subqueries. This requirement limits the answerable queries to queries having a single free variable and being decomposable, which are called tree-form queries and correspond to the $\mathcal{SROI}^-$ description logic. In this paper, we define a more general set of queries, called DAG queries and formulated in the $\mathcal{ALCOIR}$ description logic, propose a query embedding method for them, called DAGE, and a new benchmark to evaluate query embeddings on them. Given the computational graph of a DAG query, DAGE combines the possibly multiple paths between two nodes into a single path with a trainable operator that represents the intersection of relations and learns DAG-DL from tautologies. We show that it is possible to implement DAGE on top of existing query embedding methods, and we empirically measure the improvement of our method over the results of vanilla methods evaluated in tree-form queries that approximate the DAG queries of our proposed benchmark.

## Keywords

Knowledge Graph, Complex Query Answering, Description Logic

**ACM Reference Format:**

. 2024. DAGE: DAG Query Answering via Relational Combinator with Logical Constraints. In . ACM, New York, NY, USA, 16 pages. https://doi.org/10.1145/nnnnnnn.nnnnnnn

## 1 Introduction

A challenging aspect of machine learning, called *complex reasoning*, is to solve tasks that can be subdivided into subtasks. A prominent complex reasoning problem is predicting answers to queries in knowledge graphs. This problem, called *complex query answering*, involves solving queries by decomposing them into subqueries. To address this problem, several query embedding (QE) methods [1–4] encode queries with low-dimensional vectors, and utilize neural logical operators to define the embedding of a query as the combination of the embedding of its subqueries. However, these methods are only capable of processing a restricted set of first-order logic queries that have a single unquantified variable (called *target*), correspond to $\mathcal{SROI}^-$ description logic queries [5] and are called

tree-form queries because, considering only the nodes representing variables, their *computation graphs* are trees [6]. In this work, we consider a more expressive set of queries, *DAG queries*, which extends tree-form queries by allowing for quantified variables to appear multiple times in the first component of atoms. In doing so, *DAG queries* can include multiple paths from a quantified variable $x$ to a target variable $y$, whereas in tree-form queries it is at most one path from $x$ to $y$.

Consider the following first-order query $\phi(x)$, asking for works edited by an Oscar winner and produced by an Oscar winner.

$$\phi(y) ::= \exists x_1 \exists x_2 : \mathsf{wonBy}(\mathsf{Oscar}, x_1) \wedge \quad (1)$$
$$\mathsf{edited}(x_1, y) \wedge$$
$$\mathsf{wonBy}(\mathsf{Oscar}, x_2) \wedge$$
$$\mathsf{produced}(x_2, y).$$

The computation graph of query $\phi(y)$ is the following:

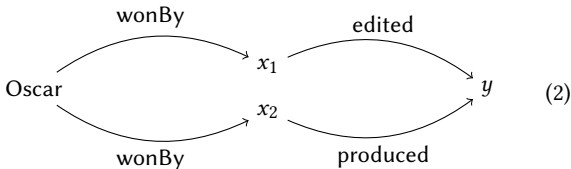

$$(2)$$

Query $\phi(y)$ is tree-form because there exists at most one path from $x_1$ to $y$ and at most one path from $x_2$ to $y$. Since it is tree-form, it can be expressed as a conjunction $\phi_1(y) \wedge \phi_2(y)$, where the subqueries $\phi_1(y)$ and $\phi_2(y)$ are also tree-form:

$$\phi_1(y) ::= \exists x_1 : \mathsf{wonBy}(\mathsf{Oscar}, x_1) \wedge \mathsf{edited}(x_1, y), \quad (3)$$
$$\phi_2(y) ::= \exists x_2 : \mathsf{wonBy}(\mathsf{Oscar}, x_2) \wedge \mathsf{produced}(x_2, y). \quad (4)$$

In $\mathcal{SROI}^-$, query $\phi$ is expressed as $C = C_1 \sqcap C_2$, where the subqueries $C_1$ and $C_2$ are:

$$C_1 ::= \exists \mathsf{edited}^-.(\exists \mathsf{wonBy}^-.\{\mathsf{Oscar}\}), \phi_2(y) \quad (5)$$
$$C_2 ::= \exists \mathsf{produced}^-.(\exists \mathsf{wonBy}^-.\{\mathsf{Oscar}\}). \quad (6)$$

Complex query answering methods use the embeddings of the subqueries $\phi_1$ and $\phi_2$ to compute the embedding of query $\phi$. To this end, these methods define a neural logical operator that represents the logical operation $\wedge$. Hence, the ability to decompose queries into subqueries and express the logical connectives with neural logical operators is critical for the existing complex reasoning methods.

Let us now show a query where this decomposition of queries does no longer hold. Consider the query asking for works edited and produced by an Oscar winner (i.e., an Oscar winner that has both roles, editor and producer). Compared with the previous query, this new query enforces $x_1 = x_2$, which can be encoded by renaming both variables $x_1$ and $x_2$ as $x$:

$$\psi(y) = \exists x : \mathsf{wonBy}(\mathsf{Oscar}, x) \wedge \quad (7)$$
$$\mathsf{edited}(x, y) \wedge$$
$$\mathsf{produced}(x, y).$$

If we observe the computation graph of query $\psi(y)$, depicted in (8), we can see that $\psi(y)$ is no longer a tree-form query because there are two paths from variable $x$ to variable $y$. We call Directed Graph Queries (DAG) to the queries in the set resulting from relaxing the maximum of one path restriction of tree-form queries.

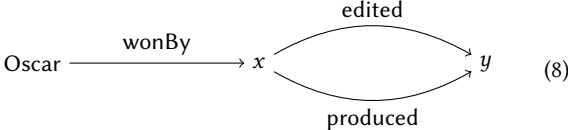

$$\text{(8)}$$

Query $\psi(y)$ cannot be decomposed into two tree-form queries because the conjunction between the query atoms $\text{edited}(x, y)$ and $\text{produced}(x, y)$ requires considering two target variables in the complex reasoning subtask. Similarly, query $\psi(y)$ is not expressible in $\mathcal{SROI}^-$ because $\mathcal{SROI}^-$ allows for conjunctions in concept descriptions but not role descriptions, which are required to indicate that $x$ "produced and edited" $y$. A description logic that allows for conjunctions in role descriptions, called $\mathcal{ALCOIR}$[1], can express the query $\psi(y)$ as the following concept description $D$[2]:

$$D ::= \exists(\text{edited} \sqcap \text{produced})^-.(\exists\text{wonBy}^-.\{\text{Oscar}\}). \quad (9)$$

Unlike existing methods, to compute the embedding of query $D$, we do not decompose $D$ into two subqueries, but we compute the embedding of the relation description $\text{edited} \sqcap \text{produced}$ with an additional neural operator, called *relational combinator*, to represent the intersection between relations. With this extension to existing methods [1–3], we can represent the aforementioned query $D$ with the following computation graph.

$$\text{Oscar} \xrightarrow{\text{wonBy}} x \xrightarrow{\text{edited} \sqcap \text{produced}} y \quad (10)$$

Like the computation graphs of tree-form queries, the computation graph of query $D$ has a single path from variable $x$ to variable $y$. Hence, we can reuse existing query embedding methods [1–3] to recursively define the embedding of a DAG query.

Without our extension, the vanilla methods cannot be applied to DAG queries. Instead, can apply them by relaxing DAG queries to tree-form queries. However, one can expect that this workaround solution produce less accurate results because the solutions to the relaxed query $\phi(y)$ in (1) are not enforced to satisfy $x_1 = x_2$ like the solutions to query $\psi(y)$ in (7).

To define the relational combinator for role conjunctions, we encourage the models to satisfy a set of well-known $\mathcal{ALCOIR}$ tautologies involving role conjunctions.

In summary, this paper makes the following contributions:

(1) In Section 3, we propose a description logic, named DAG, that extends $\mathcal{SROI}^-$ to encode conjunction of relations, and we present four tautologies involving this extension.

(2) In Section 4, we propose an integrable relational combinator that can be integrated into existing query embedding methods and generally enhance their expressiveness to DAG

---

[1] $\mathcal{ALCOIR}$ is a description logic in the family of Attributive Languages ($\mathcal{AL}$). The letters $C$, $O$, $I$, and $R$ stand for the extensions to the $\mathcal{AL}$ description logic with complement, nominals, inverse predicates, and conjunction of role descriptions [7].

[2] In this paper we use the terms concept description and query as synonyms because a concept description defines the answers to the query.

queries, and that follows three tautologies involving the intersection of relations (i.e., *commutativity*, *distributivity*, and *monotonicity*).

(3) In Section 5.1, we introduce six novel types of DAG queries, and their corresponding relaxed tree-form queries, and develop new datasets with different test difficulty levels.

(4) In Section 5.3, we assess the performance of existing methods on the created new datasets, comparing them with our integrated module. The results show that DAGE brings consistent and significant improvement to the baseline models on DAG queries.

(5) In Section 5.6, we create new data splits on the benchmark datasets to analyze DAGE's effectiveness in improving query embedding models for DAG queries in greater detail.

## 2 Preliminaries

This section presents queries as $\mathcal{ALCOIR}$ concepts. We follow the notations and semantics described in [7]. For the following definitions, we assume three pairwise disjoint sets $C$, $\mathcal{R}$, and $\mathcal{E}$, whose elements are respectively called *concept names*, *role names* and *individual names*.

*Definition 2.1 (Syntax of $\mathcal{ALCOIR}$ Concept and Role Descriptions).* $\mathcal{ALCOIR}$ concept descriptions $C$ and role descriptions $R$ are defined by the following grammar

$$C ::= \top \mid A \mid \{a\} \mid \neg C \mid C \sqcap C \mid \exists R.C$$
$$R ::= r \mid R^- \mid R \circ R \mid R \sqcap R \mid R^+$$

where the symbol $\top$ is a special concept description, and symbols $A$, $a$ and $r$ stand for concept names, individual names, and role names, respectively. Concept descriptions $\{a\}$ are called *nominals*. We write $\bot$, $C \sqcup D$, $\forall R.C$ as abbreviations for $\neg\top$, $\neg(\neg C \sqcap \neg D)$ and $\neg\exists R.\neg C$, respectively.

*Definition 2.2 (Syntax of $\mathcal{ALCOIR}$ Knowledge Bases).* Given two $\mathcal{ALCOIR}$ concept descriptions $C$ and $D$ and two role descriptions $R$ and $S$, the expressions $C \sqsubseteq D$ and $R \sqsubseteq S$ are respectively *concept inclusion* and a *role inclusion*. We write $C \equiv D$ as an abbreviation for two concept inclusions $C \sqsubseteq D$ and $D \sqsubseteq C$, and likewise for $R \equiv S$. Given two individual names $a$ and $b$, a concept description $C$ and a relation description $R$, the expression $C(a)$ is a *concept assertion* and the expression $R(a, b)$ is a *role assertion*. An $\mathcal{ALCOIR}$ knowledge base is a triple $\mathcal{K} = (\mathcal{R}, \mathcal{T}, \mathcal{A})$ where $\mathcal{R}$ is a set of role inclusions, $\mathcal{T}$ is a set of concept inclusion, and $\mathcal{A}$ is a set of concept and role assertions.

*Definition 2.3 (Interpretations).* An *interpretation* $\mathcal{I}$ is a tuple $(\Delta^{\mathcal{I}}, \cdot^{\mathcal{I}})$ where $\Delta^{\mathcal{I}}$ is a set and $\cdot^{\mathcal{I}}$ is a function with domain $\mathcal{E} \cup C \cup \mathcal{R}$, called the *interpretation function*, that maps every individual name $a \in \mathcal{E}$ to an element $a^{\mathcal{I}} \in \Delta^{\mathcal{I}}$, every concept name $A \in C$ to a set $A^{\mathcal{I}} \subseteq \Delta^{\mathcal{I}}$, and every role name $r \in \mathcal{R}$ to a relation $r^{\mathcal{I}} \subseteq \Delta^{\mathcal{I}} \times \Delta^{\mathcal{I}}$. The interpretation function is recursively extended to $\mathcal{ALCOIR}$ concept descriptions and role descriptions by defining the semantics of each connective (see [7] and Appendix B).

*Definition 2.4 (Semantics of ALCOIR Knowledge Bases).* Given an interpretation $\mathcal{I}$, we say that $\mathcal{I}$ is a *model* of

- a role axiom $R \sqsubseteq S$ if and only if $R^{\mathcal{I}} \subseteq S^{\mathcal{I}}$,

- a concept axiom $C \sqsubseteq D$ if and only if $C^{\mathcal{I}} \subseteq D^{\mathcal{I}}$,
- a concept assertion $C(a)$ if and only if $a^{\mathcal{I}} \in C^{\mathcal{I}}$,
- a role assertion $R(a, b)$ if and only if $(a^{\mathcal{I}}, b^{\mathcal{I}}) \in R^{\mathcal{I}}$,
- an $\mathcal{ALCOIR}$ knowledge base $\mathcal{K} = (\mathcal{R}, \mathcal{T}, \mathcal{A})$ if and only if $\mathcal{I}$ is a model of every element in $\mathcal{R} \cup \mathcal{T} \cup \mathcal{A}$.

*Definition 2.5 (Entailment).* Given two knowledge bases $\mathcal{K}_1$ and $\mathcal{K}_2$ we say that $\mathcal{K}_1$ *entails* $\mathcal{K}_2$, denoted $\mathcal{K}_1 \models \mathcal{K}_2$, if for every interpretation $\mathcal{I}$, if $\mathcal{I}$ models $\mathcal{K}_1$ then $\mathcal{I}$ models $\mathcal{K}_2$. This definition is extended to axioms and assertions (e.g., $\mathcal{K} \models C(a)$ if all models or $\mathcal{K}$ are also models of $C(a)$).

*Definition 2.6 (Knowledge Graph [5]).* A knowledge graph $G$ is a $\mathcal{ALCOIR}$ knowledge base whose RBox is empty, its TBox contains a unique concept incluslion $\top \sqsubseteq \{a_1\} \sqcup \cdots \sqcup \{a_n\}$, called *domain-closure assumption*, where $\{a_1, \ldots, a_n\}$ is the set of all individuals names occurring in the ABox, and its ABox contains only role assertions.

*Definition 2.7 (Knowledge Graphs Query Answers).* Given a knowledge graph $G$, the answers to an $\mathcal{ALCOIR}$ concept description $C$ are the individual names $a \in \mathcal{E}$ such that $G \models C(a)$.

## 3 Tree-Form and the DAG Queries

As was proposed by He et al. [5], tree-form queries can be expressed as $\mathcal{SROI}^-$ concepts descriptions. The computation graphs of the first-order formulas corresponding to these concepts descriptions have at most one path for every quantified variable to the target variable. As we already show, this is not hold if the relational intersection $\sqcap$ is added. In this section, we define tree-form and DAG queries as subsets of the $\mathcal{ALCOIR}$ description logic, we describe their computation graph, and the relaxation of non-tree form DAG queries as tree-form queries.

### 3.1 Syntax of Queries

*Definition 3.1 (Tree-Form and DAG queries).* DAG queries are the subset of $\mathcal{ALCOIR}$ concept descriptions $C$ defined by the following grammar

$$C ::= \{a\} \mid \neg C \mid C \sqcap C \mid \exists R.C$$
$$R ::= r \mid R^- \mid R \circ R \mid R \sqcap R$$

A DAG query is said tree-form if it does not include the operator $\sqcap$ in role descriptions.

Unlike $\mathcal{ALCOIR}$ concept descriptions, DAG queries do not include concept names, the $\top$ concept, nor the transitive closure or relations. We exclude these constructors because they are not present in queries supported by existing query embeddings.

PROPOSITION 3.2. *Given two role descriptions $R$ and $S$, and an individual name $a$, the following equivalences hold:*

commutativity: $R \sqcap S \equiv S \sqcap R$,
monotonicity: $R \sqcap S \sqsubseteq R$,
restricted conjunction preserving: $\exists (R \sqcap S).\{a\} \equiv \exists R.\{a\} \sqcap \exists S.\{a\}$.

PROOF. The tautologies follow directly from the semantics of $\mathcal{ALCOIR}$ concept and role descriptions. □

## 3.2 Computation Graphs

He et al. [5] illustrated the computation graphs for tree-form queries encoded as $\mathcal{SROI}^-$ concepts, but did not formalize them. We next provide such a formalization for a graph representation of DAG queries (and thus for tree-form queries).

*Definition 3.3 (Computation Graph).* A *computation graph* is a labelled directed graph $\Gamma = (N, E, \lambda, \tau)$ such that $N$ is a set whose elements are called *nodes*, $E \subseteq N \times N$ is a set whose elements are called *edges*, $\lambda$ is a function that maps each node in $N$ to a *label*, and $\tau$ is a distinguished node in $N$, called *target*.

*Example 3.4.* The computation graph $\Gamma = (N, E, \lambda, \tau)$ with $N = \{u_1, u_1\}$, $E = \{(u_1, u_2)\}$, $\lambda = \{u_1 \mapsto \{\text{Oscar}\}, u_2 \mapsto \exists \text{wonBy}^-\}$, and $\tau = u_2$ is depicted in (11).

$$\boxed{u_1 : \{\text{Oscar}\}}\!\!-\!\!\boxed{u_2 : \exists \text{wonBy}^-} \qquad (11)$$

Intuitively, the node $u_1$ computes the concept $\{\text{Oscar}\}$ and the node $u_2$ computes the concept $\exists \text{wonBy}^-.\{\text{Oscar}\}$, which corresponds to answers to the query asking who is an Oscar's winner.

To define the computation graphs of DAG queries, we need to introduce the composition of a computation graph $\Gamma$ with a role description $R$, denoted $\Gamma[R]$. Intuitively, the composition is the concatenation of $\Gamma$ with the graph representing the role description, as Example 3.5 illustrates. A definition for this operation and the computation graph of DAG queries is suplemented in Appendix C.

*Example 3.5.* Consider the computation graph $\Gamma$ depicted in (11). The computation graph $\Gamma[\text{edited}^-]$ and $\Gamma[(\text{edited} \sqcap \text{produced})^-]$ are depicted in (12) and (13), respectively.

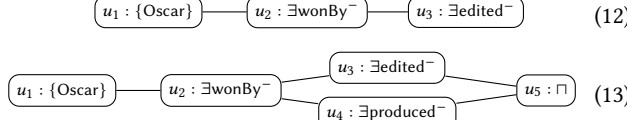

$$(12)$$

$$(13)$$

*Example 3.6.* Consider the tree-form query $C = C_1 \sqcap C_2$ defined by equations (5) and (6). The computation graph of $C$ is depicted in the following diagram:

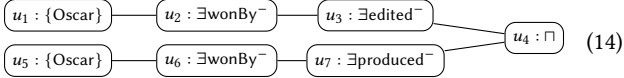

$$(14)$$

Similarly, the computation graph for the DAG query in equation (9) is depicted by the figure in (13).

Notice that, in (14), different nodes can have the same label and represent the same concept (e.g., nodes $u_2$ and $u_6$ represent the concept $\exists \text{wonBy}^-.\{\text{Oscar}\}$). Intuitively, the duplication of labels means that an answer can be a work edited by an Oscar's winner and produced by another Oscar's winner. On the other hand, since there is a single node for this concept in the computation graph in, (13), namely node $u_2$, the work must be produced and edited by the same Oscar's winner.

## 3.3 Relaxing Non Tree-Form DAG queries

The restricted conjunction preserving (see Proposition 3.2), does no longer follow if we replace the nominal $\{a\}$ with a general concept description $C$. Indeed, the example discussed in the introduction is

a counterexample for the generalized version of the conjunction preserving. The fact that conjunction is not preserved in general is the cause of the need of new neural operator for the role conjunction, different from the one used for the concept conjunction. Alternatively, if the neural operator is not used, we can relax concept description with a relaxed version of this tautology.

PROPOSITION 3.7. *Given two role descriptions $R$ and $S$, and a concept description $C$, the following concept inclusion hols:*

$$\exists(R \sqcap S).C \sqsubseteq \exists R.C \sqcap S.C \tag{15}$$

PROOF. By monotonicity, concept $\exists(R \sqcap S).C$ is included in the concepts $\exists R.C$ and $\exists S.C$. Then, concept $\exists(R \sqcap S).C$ is included in the concept $\exists R.C \sqcap \exists S.C$. □

Intuitively, the role conjunction relaxation consists of not assuming that the instances of the concept $C$ must be equal on the concept defined on the right side. An example of this was discussed in the introduction, when the editor and producer of a work are not required to be the same Oscar's winner. Thus, the three-form query with the computation graph in (14) relaxes the non tree-form with the computation graph in (13).

*Definition 3.8 (Tree-form approximation).* The *approximated tree-form query* of a DAG query $Q$, denoted $\text{tree}(Q)$, is the tree-form query resulting from removing every conjunction of role descriptions using the inclusion in (15).

It is not difficult to see that for every DAG query $Q$ with no complement constructor (i.e., without $\neg$), it holds that $Q \sqcap \text{tree}(Q)$, that is, query $\text{tree}(Q)$ relaxes query $Q$. This is not necessary for queries including complement because they are not necessarily monotonic. Hence, query embeddings that use tree-form queries to predict answers to DAG queries are expected to incur in both, more false positives and more false negatives.

## 4 DAG Query Answering with Relational Combinator

In this section, we first introduce a generalized query embedding model subsuming various previous query embedding approaches [1–3]. Then, we introduce a relational combinator that extends existing query embeddings to support DAG query type. Finally, we discuss how to introduce additional logical constraints to further improve the results.

### 4.1 Base Query Embedding Methods Interface

Many query embedding methods [1–3] predict query answers by comparing the embedding of individuals with the embedding of the query, so the individuals that are closer to the query in the embedding space are more likely to be answers. These query embeddings are learnable parameterized objects and are computed via neural operations that correspond to the logic connectives in the queries. In this subsection we present the interface required for the embedding methods to be used as a base for our proposed query embedding method, DAG-E. Query embedding methods such as Query2Box, BetaE, and ConE satisfy this interface.

We assume three vector spaces $\mathbf{E}^d$, $\mathbf{R}^d$, and $\mathbf{Q}^d$, where $\mathbf{E}$, $\mathbf{R}$, and $\mathbf{Q}$ are fields (which depend on the query embedding method) and $d$

is the dimension of the vectors. We assume that every individual $a$ is embedded in a vector $\mathbf{Emb}_a \in \mathbf{E}^d$, every role name $r$ and its inverse $r^-$ are embedded in vector $\mathbf{Emb}_r, \mathbf{Emb}_{r^-} \in \mathbf{R}^d$, and every tree-form query $Q$ is embedded in a vector $\mathbf{Emb}_Q \in \mathbf{Q}^d$. Whereas the embedding function $\mathbf{Emb}.$ is defined for individuals and role names and the inverse of role names, because they are directly defined by the parameters to be learn, function $\mathbf{Emb}.$ is not directly defined for compound role and concept descriptions.

*4.1.1 Role Embeddings.* The embedding of a role description $R$ is recursively computed from the embedding of role names and its inverses as with a neural operators with signature RComposition : $\mathbf{R}^d \times \mathbf{R}^d \rightarrow \mathbf{R}^d$ as follows:

$$\mathbf{Emb}_{R \circ S} ::= \text{RComposition}(\mathbf{Emb}_R, \mathbf{Emb}_S), \tag{16}$$
$$\mathbf{Emb}_{R^{--}} ::= \mathbf{Emb}_R, \tag{17}$$
$$\mathbf{Emb}_{(R \circ S)^-} ::= \mathbf{Emb}_{S^- \circ R^-}. \tag{18}$$

*4.1.2 Concept Embeddings.* The embedding of a query is computed from the embedding of individual names and role names using neural operators that represent the logical connectives in queries. The signatures of these neural operators are the following: Nominal : $\mathbf{E}^d \rightarrow \mathbf{Q}^d$, RelT : $\mathbf{Q}^D \times \mathbf{R}^E \rightarrow \mathbf{Q}^E$, Intersect : $\mathbf{Q}^d \times \ldots \mathbf{Q}^d \rightarrow \mathbf{Q}^d$, and Complement : $\mathbf{Q}^d \rightarrow \mathbf{Q}^d$. These neural operators define query embedding of tree-form queries as follows:

$$\mathbf{Emb}_{\{a\}} ::= \text{Nominal}(\mathbf{Emb}_a), \tag{19}$$
$$\mathbf{Emb}_{\exists r.C} ::= \text{RelT}_r(\mathbf{Emb}_C), \tag{20}$$
$$\mathbf{Emb}_{C_1 \sqcap C_2 \ldots \sqcap C_n} ::= \text{Intersect}(\mathbf{Emb}_{C_1}, \mathbf{Emb}_{C_2}, \cdots, \mathbf{Emb}_{C_n}), \tag{21}$$
$$\mathbf{Emb}_{\neg C} ::= \text{Complement}(\mathbf{Emb}_C). \tag{22}$$

*4.1.3 Insideness.* Given a query $Q$ and a knowledge graph $G$, the goal of query embedding approaches is to maximize the predictions of *positive answers* to query $Q$ (i.e., individuals $a$ such that $G \models Q(a)$) and minimize the prediction of *negative answers* to query $Q$ (i.e., individuals $b$ such that $G \models \neg Q(b)$). Because of the open-world semantics of $G$ (see Definition 2.6) we cannot know which answers are negative. However, the learning of query embedding needs negative answers. Therefore, for each positive answer $a$, query embedding methods assume a random individual $b$, different from $a$, to be a negative answer.

In the representation space, the evaluation of how likely an individual is an answer to a query is computed with a function with signature Insideness : $\mathbf{Q}^d \times \mathbf{Q}^d \rightarrow \mathbb{R}$, that returns higher numbers for individuals that are answers to the query than for individuals that are not answers to the query. That is, given a query $Q$ with a positive answer $a$ and its corresponding randomly generated negative answer $a'$ distinct from $a$, the goal of query embedding approaches is to minimize the following loss:

$$\mathcal{L}_i(Q) ::= \sum_{a \in \mathcal{E}} \left( \begin{array}{l} -\log \sigma \left( \gamma - \text{Insideness}(\mathbf{Emb}_Q, \mathbf{Emb}_{\{a\}}) \right) \\ + \sum_{j}^{k} \frac{1}{k} \log \sigma \left( \gamma - \text{Insideness}(\mathbf{Emb}_Q, \mathbf{Emb}_{\{a'\}}) \right) \end{array} \right). \tag{23}$$

where $\{a'\}$ us the negative sample, $\gamma$ is a margin hyperparameter and $k$ is the number of random negative samples for each positive query answer pair.

## 4.2 The Relational Combinator

So far, we have described an interface consisting of neural operators that are implemented by existing query embeddings. These neural operators allow the computation of tree-form query embeddings, but not not DAG queries including the conjunction of roles. To enhance the capability of these methods for DAG queries, we introduce a relational combination operator with signature

$$\text{RCombiner}_k : (\mathbf{R}^d)^k \rightarrow \mathbf{R}^d, \tag{24}$$

where $k$ is a positive natural number. The embedding of a role description $R_1 \sqcap \cdots \sqcap R_k$ (with $k > 0$) is:

$$\mathbf{Emb}_{R_1 \sqcap \cdots \sqcap R_k} ::= \text{RCombiner}_k(\mathbf{Emb}_{R_1}, \ldots, \mathbf{Emb}_{R_k}), \tag{25}$$

where RCombiner is a commutative neural network. We used the neural operator DeepSet [8] to implement RCombiner.

$$\text{RCombiner}_k(\mathbf{Emb}_{R_1}, \ldots, \mathbf{Emb}_{R_k}) = \sum_{1 \le i \le k} \alpha_i \cdot \text{MLP}(\mathbf{Emb}_{R_i})) \tag{26}$$

where the weights $\alpha_1, \cdots, \alpha_k$ sum 1. Specifically,

$$\alpha_i = \frac{\exp(\text{MLP}(\mathbf{Emb}_{R_i}))}{\sum_{1 \le j \le k} \exp(\text{MLP}(\mathbf{Emb}_{R_j}))}.$$

**PROPOSITION 4.1.** *Given two role descriptions $R$ and $S$,*

$$\text{RCombiner}_2(\mathbf{Emb}_R, \mathbf{Emb}_S) = \text{RCombiner}_2(\mathbf{Emb}_S, \mathbf{Emb}_R). \tag{27}$$

**PROOF.** It follows from the commutativity of the DeepSet. □

Proposition 4.1 guarantees that the embedding of role description satisfies some of the $\mathcal{ALCOIR}$ tautologies described in Proposition 3.2, namely commutativity and idempotence.

## 4.3 Encouraging Tautologies

So far, we have shown (see Proposition 4.1) that the proposed relational combinator satisfies two of the $\mathcal{ALCOIR}$ tautologies presented in Proposition 3.2, namely commutative and idempotence, but not the monotonicity and the restricted conjunction preserving. We hypothesize that by encouraging the query embeddings such that the inference of embeddings follow these tautologies, we can improve the embedding generalization capacity.

*4.3.1 Monotonicity.* We encourage the query embeddings of a DAG query $Q = \exists(R \sqcap S).C$ to be subsumed by the query embedding of query $Q' = \exists R.C$ by introducing the following loss:

$$\mathcal{L}_m(Q) = \sum_{r \in \mathcal{R}, s \in \mathcal{R}} \text{Insideness}(\mathbf{Emb}_Q, \mathbf{Emb}_{Q'}), \tag{28}$$

Intuitively, Insideness measures the likelihood of $\mathbf{Emb}_{Q'}$ being inside $\mathbf{Emb}_Q$.[3]

---

[3]More details about the Insideness function for individual methods can be found in Appendix A.

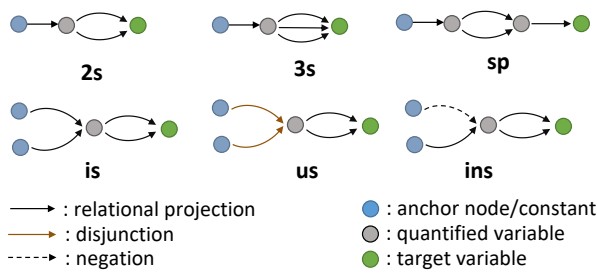

**Figure 1: Query structures considered in the experiments, where anchor entities and relations are to be specified to instantiate logical queries. Naming for each query structure is provided under each subfigure, where "s", "p", "i", "u" and "n" stand for "split", "projection", "intersection", "union", and "negation" respectively. For example, "2s" stands for 2 splitting edges in the query structure.**

*4.3.2 Restricted conjunction preserving.* We encourage the tautology $\exists(r \sqcap s).\{e\} \equiv \exists r.\{e\} \sqcap \exists s.\{e\}$ (see Proposition 3.2) with the following loss:

$$\mathcal{L}_r ::= \text{Diff}(\mathbf{Emb}_{\exists(r \sqcap s).\{e\}}, \text{Intersect}(\mathbf{Emb}_{r.\{e\}}, \mathbf{Emb}_{s.\{e\}})), \tag{29}$$

where Diff measures the distance between two query embeddings. We supplement the details on Diff of each individual query embedding method in Appendix A.

By imposing these loss terms, the tautologies are encoded into geometric constraints, which are soft constraints over the embedding space. Hence, our loss terms can also be viewed as regulation terms that reduce the embedding search space. Given a query answering training dataset $\mathcal{D}$, our final optimization objective is:

$$\mathcal{L}(D) ::= \sum_{i=1}^{|\mathcal{D}|} \mathcal{L}_i(Q) + \lambda_1 \mathcal{L}_m + \lambda_2 \mathcal{L}_r, \tag{30}$$

where $\mathcal{L}_q$ is the query embedding loss, and $\lambda_1$ and $\lambda_2$ are the weights of regularization terms.

## 5 Experiments

In this section, we answer the following research questions with experimental statistics and corresponding case analyses. **RQ1:** How effectively does DAGE enhance the existing baselines in discovering answers to DAG queries that cannot be found by simply traversing the incomplete KG? **RQ2:** How well does the existing baselines with DAGE perform on tree-form queries? **RQ3:** How do the logical constraints influence the performance of DAGE? All experiments and the data generation codes are available via https://anonymous.4open.science/r/DAG_RC-9B18/README.md.

## 5.1 DAG Query Generation

Existing datasets, e.g. NELL-QA, FB15k237-QA, WN18RR-QA, do not contain DAG queries. We propose six new DAG query types, i.e., 2s, 3s, sp, us, is, and ins, as shown in Figure 1. Following these new query structures, we generate new DAG query benchmark datasets, NELL-DAG, FB15k-237-DAG, and FB15k-DAG.

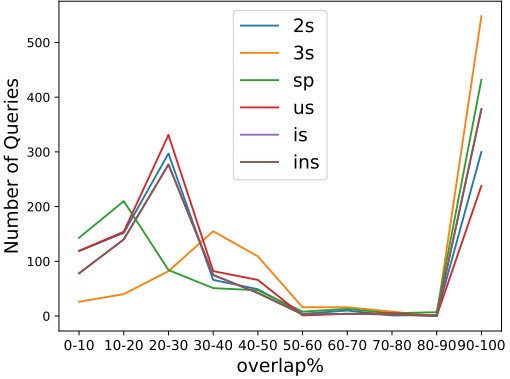

**Figure 2: Proportion of overlap between DAG query answers and the corresponding Tree-Form query answers from NELL-DAG Easy test dataset.**

Given that the answers to DAG queries represent a subset of those derived from the relaxed tree-form queries, evaluating the model's ability to handle DAG queries becomes challenging when there exists a high proportion of overlap between the answer sets of both queries. Specifically, a query embedding method can relax a DAG query into tree-form query by replacing all $\exists r \sqcap t.C$ with $\exists r.C \sqcap \exists t.C$.[4] This method could still achieve good performance if the answer set of this DAG query overlaps highly with that of the tree-form query. To validate this hypothesis in our datasets, we analyze the randomly generated DAG queries from NELL [9]. Figure 2 illustrates that around 50% of these DAG queries have answer sets that highly overlap (over 90%) with the answer sets of tree-form queries. Same analyses on the randomly generated DAG queries from FB15k and FB15k-237 can be found in Appendix E.

To solve this problem, we propose test datasets of two difficulty levels for each benchmark DAG-QA dataset in Table 6, **test-easy** and **test-hard**, such that

- **Test-easy** datasets are randomly generated, and the answer sets of some queries are probably highly similar to those of the corresponding tree queries.
- **Test-hard** datasets are selected out of the random queries such that the overlapping ratio between the answer sets of these queries and their corresponding tree queries is less than 0.5. For example, if the answer set of a DAG query is $\{a, b, c\}$ and that of its corresponding tree query is $\{a, b, c, d\}$ then this DAG query should be dismissed because the overlap ratio is 3/4.

### 5.2 Experimental Setup

**Evaluation Metrics.** We use Mean Reciprocal Rank (MRR) as the evaluation metric. Given a query $Q$, MRR represents the average of the reciprocal ranks of results, MRR $= \frac{1}{|Q|} \sum_{i=1}^{|Q|} \frac{1}{\text{rank}_i}$.

**Hyperparameters and Computational Resources.** All of our experiments are implemented in Pytorch [10] framework and run on four Nvidia A100 GPU cards. For hyperparameters search, we

---

[4]We supplement the relaxed tree-form query types corresponding to proposed DAG types in Appendix D.

performed a grid search of learning rates, the batch size, the negative sample sizes, the regularization weights $\omega$ and the margin $\gamma$. Further experimental details and the best hyperparameters are shown in Table 7 in Appendix F.

### 5.3 RQ1: How effective is DAGE for enhancing baseline models on DAG queries?

To assess the performance of DAGE in extending tree-form query embedding methods to DAG queries, we conducted the following analysis. Firstly, we retrained and tested the baseline models, Query2Box[1], BetaE[3] and ConE[2], on the new benchmark datasets by decomposing the DAG queries into the conjunction of tree-form queries. More details about the implementation of baselines are elaborated in Appendix A. Then, we implement DAGE on top of these methods and evaluate them on the new benchmark datasests again under both easy and hard test modes.

***Main Results:*** Tables 1 and 2 summarize the performance of the baseline methods, both with and without DAGE, under the easy and hard test modes. Based on these results, we draw the following conclusions. Firstly, the baseline methods show a significant performance drop from the easy to hard datasets due to the exclusion of "easy" DAG queries, as described in Section 5.1. This highlights the importance of developing datasets that effectively assess a model's ability to handle DAG queries, rather than just tree-form queries. Secondly, DAGE consistently delivers significant improvements to all baseline methods across all query types and datasets, in both easy and hard test modes. Specifically, DAGE significantly improves performance on the NELL-DAG dataset, with the average accuracy of baseline models nearly doubling when combined with DAGE compared to their standalone performance. Beyond these baseline models, we also implement other query embedding models, e.g., CQD[11] and BiQE[4], that are theoretically believed to be capable of handling DAG queries, on our new datasets. We perform comparison between the enhanced baselines and these methods in Appendix H. It demonstrates that DAGE can easily extend existing tree-form query embedding models to outperform these methods on DAG query datasets, further reinforcing its effectiveness.

### 5.4 RQ2: How well does DAGE perform on the existing benchmarks with tree-form queries?

An effective method for extending existing query embedding techniques to handle DAG queries should also ensure strong performance on the tree-form queries these methods were originally designed to process. Table 3 presents the performance of the baseline models, as well as their performance when integrated with DAGE, on tree-form queries from NELL-QA [3]. The complete results on other two datasets are supplemented in Appendix G. DAGE enables these models to handle DAG queries while preserving their original performance on tree-form queries. More importantly, DAGE shows significant improvement only on DAG queries, with little effect on tree-form queries, supporting our assumption that DAGE effectively enhances baseline performance for the new DAG query types.

**Table 1: The MRR performance of baseline models and our proposed version DAGE on easy benchmark datasets.**

| Dataset | Model | 2s | 3s | sp | is | us | $Avg_{nn}$ | ins | Avg |
|---|---|---|---|---|---|---|---|---|---|
| NELL-DAG | Query2Box | 20.53 | 34.03 | 0.10 | 23.31 | 28.2 | 21.23 | - | - |
| | Query2Box (DAGE) | 37.61 ↑ | 49.42 ↑ | **41.71** ↑ | **40.57** ↑ | **42.75** ↑ | 42.41 ↑ | - | - |
| | BetaE | 15.30 | 28.78 | 13.10 | 17.72 | 16.69 | 18.32 | 27.30 | 19.82 |
| | BetaE (DAGE) | 36.87 ↑ | 57.14 ↑ | 34.95 ↑ | 39.90 ↑ | 37.80 ↑ | 41.33 ↑ | **34.68** ↑ | **40.22** ↑ |
| | ConE | 23.55 | 39.38 | 19.48 | 25.28 | 25.01 | 26.54 | 27.71 | 26.73 |
| | ConE (DAGE) | 33.50 ↑ | **57.35** ↑ | 38.43 ↑ | 37.93 ↑ | 33.74 ↑ | 40.19 ↑ | 33.94 ↑ | 39.15 ↑ |
| FB15k-237-DAG | Query2Box | 6.84 | 11.61 | 9.26 | 6.48 | 4.61 | 7.76 | - | - |
| | Query2Box (DAGE) | **7.41** ↑ | **12.64** ↑ | 10.07 ↑ | **7.32** ↑ | **5.03** ↑ | 8.49 ↑ | - | - |
| | BetaE | 4.81 | 8.17 | 7.52 | 5.0 | 2.71 | 5.64 | 4.49 | 5.45 |
| | BetaE (DAGE) | 6.27 ↑ | 12.11 ↑ | 9.64 ↑ | 6.66 ↑ | 4.09 ↑ | 7.75 ↑ | **6.58** ↑ | 7.56 ↑ |
| | ConE | 4.90 | 9.21 | 8.88 | 5.52 | 3.08 | 6.32 | 4.80 | 6.06 |
| | ConE (DAGE) | 6.87 ↑ | 11.66 ↑ | **12.36** ↑ | 6.90 ↑ | 4.80 ↑ | **9.54** ↑ | 6.08 ↑ | **8.12** ↑ |
| FB15k-DAG | Query2Box | 32.62 | 35.52 | 20.90 | 27.79 | 23.94 | 28.15 | - | - |
| | Query2Box (DAGE) | 37.74 ↑ | 42.93 ↑ | 24.30 ↑ | 29.37 ↑ | 25.97 ↑ | 31.46 ↑ | - | - |
| | BetaE | 25.91 | 33.13 | 28.20 | 22.21 | 23.31 | 26.55 | 19.02 | 25.29 |
| | BetaE (DAGE) | 32.65 ↑ | 46.17 ↑ | 32.48 ↑ | 28.15 ↑ | 28.10 ↑ | 33.50 ↑ | 25.39 ↑ | 32.15 ↑ |
| | ConE | 32.10 | 37.42 | 32.37 | 27.14 | 27.85 | 31.37 | 23.48 | 30.06 |
| | ConE (DAGE) | **41.67** ↑ | **56.70** ↑ | **33.36** ↑ | **36.54** ↑ | **32.36** ↑ | **40.12** ↑ | **30.86** ↑ | **38.58** ↑ |

**Table 2: The MRR performance of baseline models and our proposed version DAGE on hard benchmark datasets.**

| Dataset | Model | 2s | 3s | sp | is | us | $Avg_{nn}$ | ins | Avg |
|---|---|---|---|---|---|---|---|---|---|
| NELL-DAG | Query2Box | 7.47 | 5.19 | 0.11 | 8.54 | 12.28 | 6.72 | - | - |
| | Query2Box (DAGE) | 25.38 ↑ | 20.13 ↑ | 21.25 ↑ | 24.85 ↑ | 29.24 ↑ | 24.17 ↑ | - | - |
| | BetaE | 14.38 | 16.23 | 7.99 | 13.32 | 13.03 | 12.99 | 28.17 | 15.52 |
| | BetaE (DAGE) | 27.68 ↑ | 32.25 ↑ | 16.36 ↑ | 26.14 ↑ | 29.19 ↑ | 26.32 ↑ | 33.64 ↑ | 27.54 ↑ |
| | ConE | 20.31 | 18.88 | 12.02 | 19.59 | 22.31 | 18.62 | 29.45 | 20.43 |
| | ConE (DAGE) | **30.71** ↑ | **38.41** ↑ | **24.76** ↑ | **28.44** ↑ | **31.06** ↑ | **30.67** ↑ | 34.07 | **31.24** ↑ |
| FB15k-237-DAG | Query2Box | 4.25 | 2.64 | 7.21 | 4.56 | 3.63 | 4.45 | - | - |
| | Query2Box (DAGE) | 4.81 ↑ | **2.81** ↑ | 7.87 ↑ | **5.26** ↑ | **4.38** ↑ | **6.95** ↑ | - | - |
| | BetaE | 3.62 | 1.62 | 6.44 | 3.85 | 2.42 | 3.20 | 4.31 | 3.38 |
| | BetaE (DAGE) | **4.89** ↑ | 1.66 ↑ | 8.28 ↑ | 4.75 ↑ | 3.50 ↑ | 4.61 ↑ | **6.06** ↑ | 4.85↑ |
| | ConE | 3.48 | 2.28 | 7.36 | 4.23 | 2.92 | 4.05 | 4.65 | 4.15 |
| | ConE (DAGE) | 4.78 ↑ | 2.09 | **9.72** ↑ | 4.84 ↑ | 4.16 ↑ | 5.12 ↑ | 5.25 ↑ | **5.14** ↑ |
| FB15k-DAG | Query2Box | 31.86 | 33.32 | 18.46 | 25.59 | 22.59 | 26.36 | - | - |
| | Query2Box (DAGE) | 33.78 ↑ | 39.67 ↑ | 19.61 ↑ | 26.91 ↑ | 24.76 ↑ | 28.95 ↑ | - | - |
| | BetaE | 24.02 | 31.82 | 26.12 | 20.17 | 21.93 | 24.81 | 18.60 | 23.77 |
| | BetaE (DAGE) | 30.57 ↑ | 44.30 ↑ | 29.35 ↑ | 25.72 ↑ | 26.63 ↑ | 31.31 ↑ | 25.18 ↑ | 30.29 ↑ |
| | ConE | 30.42 | 36.29 | **30.46** | 25.67 | 27.14 | 29.99 | 22.66 | 28.77 |
| | ConE (DAGE) | **40.14** ↑ | **57.06** ↑ | 29.23 | **34.63** ↑ | **31.45** ↑ | **38.50** ↑ | **30.74** ↑ | **37.21** ↑ |

**Table 3: The MRR performance of the retrained baseline models with DAGE method on tree-form query benchmark datasets**

| Dataset | Model | 1p | 2p | 3p | 2i | 3i | pi | ip | 2u | up | Avg |
|---|---|---|---|---|---|---|---|---|---|---|---|
| NELL-QA | Query2Box | 42.7 | 14.5 | 11.7 | 34.7 | 45.8 | 23.2 | 17.4 | 12.0 | 10.7 | 23.6 |
| | Query2Box (DAGE) | 42.1 | 23.4 | 21.3 | 28.6 | 41.1 | 20.0 | 12.3 | 27.5 | 15.9 | 28.3 |
| | BetaE | 53.0 | 13.0 | 11.4 | 37.6 | 47.5 | 24.1 | 14.3 | 12.2 | 8.5 | 24.6 |
| | BetaE (DAGE) | 53.4 | 12.9 | 10.8 | 37.6 | 47.1 | 23.8 | 13.8 | 12.3 | 8.3 | 24.4 |
| | ConE | 53.1 | 16.1 | 13.9 | 40.0 | 50.8 | 26.3 | 17.5 | 15.3 | 11.3 | 27.2 |
| | ConE (DAGE) | 53.2 | 15.7 | 13.7 | 39.9 | 50.7 | 26.0 | 17.0 | 14.8 | 10.9 | 26.8 |

## 5.5 RQ3: How do the logical constraints influence the performance of DAGE?

Table 4 summarizes the performances of baseline models enhanced by DAGE with additional logical constraints, i.e., monoticity and restricted conjunction preservation in proposition 3.2. First, we find that both monoticity and restricted conjunction preservation bring some improvements in general. The improvements of the monotic­ity regularization is bigger than that of the restricted conjunction preservation. Next, we find that the combination of both logical constraints consistently enhances DAGE's performance on DAG

**Table 4: The MRR performance of baseline models with DAGE on NELL-DAG hard benchmark dataset, along with their performance when integrated with additional logical constraints.**

| Model | 2s | 3s | sp | is | us | Avg$_{nn}$ | ins | Avg |
|---|---|---|---|---|---|---|---|---|
| Query2Box (DAGE) | 25.38 | 20.13 | 21.25 | 24.85 | 29.24 | 24.17 | - | - |
| Query2Box (DAGE+Distr) | **25.93** | 21.50 | 20.85 | 24.73 | 29.71 | 24.54 | - | - |
| Query2Box (DAGE+Mono) | 25.90 | 21.74 | 21.87 | 25.41 | **30.17** | 25.02 | - | - |
| Query2Box (DAGE+Distr+Mono) | 25.87 | **22.01** | 22.34 | 24.96 | 30.02 | **25.04** | - | - |
| BetaE (DAGE) | 27.68 | 32.25 | 16.36 | 26.14 | 29.19 | 26.32 | 33.64 | 27.54 |
| BetaE (DAGE+Distr) | 27.91 | 32.87 | 17.12 | **27.02** | **30.13** | 27.01 | 34.28 | 28.22 |
| BetaE (DAGE+Mono) | 28.01 | **33.56** | 16.89 | 26.93 | 29.47 | 26.97 | 34.17 | 28.17 |
| BetaE (DAGE+Distr+Mono) | **28.11** | 33.48 | **17.67** | 26.83 | 29.36 | **27.09** | **34.49** | **28.32** |
| ConE (DAGE) | 30.71 | 38.41 | 24.76 | 28.44 | 31.06 | 30.67 | 34.07 | 31.24 |
| ConE (DAGE+Distr) | 31.23 | 39.37 | 25.08 | **28.73** | **31.92** | 31.27 | **36.34** | 32.11 |
| ConE (DAGE+Mono) | 31.47 | 39.82 | 25.17 | 28.57 | 31.24 | 31.25 | 35.84 | 32.02 |
| ConE (DAGE+Distr+Mono) | **31.88** | **39.89** | 25.28 | 28.64 | 31.57 | **31.45** | 35.93 | **32.20** |

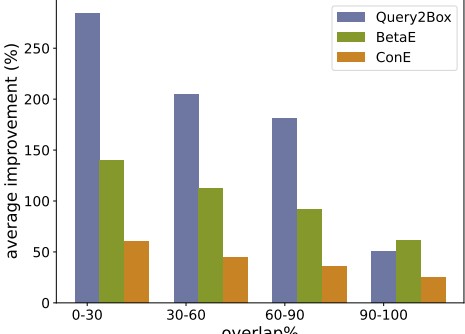

**Figure 3: Average percentage of improvement from baseline model to baseline (DAGE) for different overlap ratios**

query answering tasks, highlighting the importance of incorporating constraints in complex query answering.

## 5.6 Ablation study

To more effectively examine the specific impact of DAGE on baseline models to DAG queries, a detailed analysis was conducted on them based on the NELL-DAG query answering dataset. We divide the easy test dataset into four groups, $0 - 30\%$, $30 - 60\%$, $60 - 90\%$, and $90 - 100\%$, based on the overlap ratio between DAG query answers and their corresponding tree-form query answers. The specific number of queries in each category can be found in Figure 2. Figure 3 shows the average performance improvement, in percentage, of the baseline models when enhanced with DAGE across the subgroups of test queries. It is observed that most of DAGE's improvements occur in queries with lower answer overlap ratios. For queries with higher overlap ratios, which are easier for the baseline models, DAGE bring less improvement. This shows that DAGE significantly improves baseline models, particularly on challenging tasks they previously couldn't handle on their own.

## 6 Related Work

*Query Embedding Methods.* Path-based [12, 13], neural [2–4, 14, 15], and neural-symbolic [11, 16, 17] methods have been developed to answer (subsets of) queries. Among these methods, geometric and probabilistic query embedding approaches [2, 3, 14, 15] provide an effective way to answer tree-form queries over incomplete and noisy KGs. These methods achieve this by representing sets of entities as geometric shapes or probability distributions, such as boxes [15], cones [2], or Beta distributions [3], and applying neural logic operations directly on these representations. The Graph Query Embedding (GQE) method [14] was one of the earliest approaches, designed to handle only conjunctive queries by representing the query $q$ as a single vector using neural translational operations. However, representing a query as a single vector limits its ability to effectively capture multiple entities. Query2Box [15] addresses this limitation by representing entities as points within boxes, enabling it to model the intersection of entity sets as the intersection of boxes in vector space. ConE [2] introduced the first geometry-based query embedding approach capable of handling negation by embedding entity sets (or query embeddings) as cones in Euclidean space. However, these established theories and methods are limited to tree-form queries. There is a lack of techniques that can extend their application to DAG queries.

## 7 Conclusion

In this paper, we define a more general set of queries, called DAG queries and discuss its connection to $\mathcal{ALCOIR}$ description logic. We propose DAGE, a plug-and-play module that extends existing tree-form query embedding approaches to handle DAG queries, whose computation graphs contain more than one paths between two nodes. DAGE handle this issue by merging the possible multiple paths through a relational combinator, which corresponds to the conjunction operator of relations in $\mathcal{ALCOIR}$). We propose proper regularization terms to encourage the inference of query embeddings to satisfy desired tautologies including monotonicity and restricted conjunction preserving. We create novel benchmarks consisting of DAG queries for evaluating DAG query embedding approaches. We implementDAGE upon three existing query embedding approaches, and the results show that DAGE significantly outperforms its corresponding counterpart on DAG queries while maintaining competitive performance on tree-form queries.

One limitation of DAGE is that it does not enforce hard constraints over tautologies, as in practice the regularization loss cannot be zero. In future work, we will explore embedding approaches that directly respect these tautologies without regularization terms.

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

# A Specific details of computations of baseline models with DAGE

In this section, we supplement the specific details of the computations, i.e., relational transformation, intersection operator and complement operator, of the query embedding models involved in our experiments. Note that we do not introduce the union operator because queries involving with union can be translated into disjunctive normal form (DNF), more details can be checked in [15].

## A.1 Query2Box

Query2Box models concepts in the vector space using boxes (i.e., axis-aligned hyper-rectangles) and defines a box in $\mathbb{R}^d$ by $\mathbf{p} = (\text{Cen}(\mathbf{p}), \text{Off}(\mathbf{p})) \in \mathbb{R}^{2d}$ as

$$\text{Box}_\mathbf{p} \equiv \left\{ \mathbf{v} \in \mathbb{R}^d : \text{Cen}(\mathbf{p}) - \text{Off}(\mathbf{p}) \leq \mathbf{v} \leq \text{Cen}(\mathbf{p}) + \text{Off}(\mathbf{p}) \right\} \quad (31)$$

where $\leq$ is element-wise inequality, $\text{Cen}(\mathbf{p}) \in \mathbb{R}^d$ is the center of the box, and $\text{Off}(\mathbf{p}) \in \mathbb{R}^d_{\geq 0}$ is the positive offset of the box, modeling the size of the box.

The operations of concepts can be defined by

- **Relational Transformation** maps from one box to another box using a box-to-box translation. This is achieved by translating the center and getting a larger offset. This is modeled by $\mathbf{p} + \mathbf{r}$, where each relation r is associated with a relation embedding $\mathbf{r} = (\text{Cen}(\mathbf{r}), \text{Off}(\mathbf{r})) \in \mathbb{R}^{2d}$.
- **Intersection Operator** models the intersection of a set of box embedding $\{\mathbf{p}_1, \cdots, \mathbf{p}_n\}$ as $\mathbf{p}_{inter} = (\text{Cen}(\mathbf{p}_{inter}), \text{Off}(\mathbf{p}_{inter}))$, such that

$$\text{Cen}(\mathbf{p}_{inter}) = \sum_i \mathbf{w}_i \odot \text{Cen}(\mathbf{p_i}), \quad (32)$$

where

$$\mathbf{w}_i = \frac{\exp(\text{MLP}(\mathbf{p_i}))}{\sum_j \exp(\text{MLP}(\mathbf{p_j}))},$$

$$\begin{aligned} \text{Off}(\mathbf{p}_{inter}) = \text{Min}(\{\text{Off}(\mathbf{p_1}), \ldots, \text{Off}(\mathbf{p_n})\}) \\ \odot \sigma(\text{DeepSets}(\{\mathbf{p_1}, \ldots, \mathbf{p_n}\})), \end{aligned} \quad (33)$$

where $\odot$ is the dimension-wise product, $\text{MLP}(\cdot) : \mathbb{R}^{2d} \rightarrow \mathbb{R}^d$ is the Multi-Layer Perceptron, $\sigma(\cdot)$ is the sigmoid function, $\text{DeepSets}(\cdot)$ is the permutation-invariant deep architecture [8], and both $\text{Min}(\cdot)$ and $\exp(\cdot)$ are applied in a dimension-wise manner.
- **Distance function** Given a query box $\mathbf{p} \in \mathbb{R}^{2d}$ and an entity vector $\mathbf{a} \in \mathbb{R}^d$, their distance is define as

$$\text{dist}_{box}(\mathbf{a}; \mathbf{p}) = \text{dist}_{outside}(\mathbf{a}; \mathbf{p}) + \alpha \cdot \text{dist}_{inside}(\mathbf{a}; \mathbf{p}), \quad (34)$$

where $\mathbf{q}_{max} = \text{Cen}(\mathbf{p}) + \text{Off}(\mathbf{p}) \in \mathbb{R}^d$, $\mathbf{p}_{min} = \text{Cen}(\mathbf{p}) - \text{Off}(\mathbf{p}) \in \mathbb{R}^d$ and $0 < \alpha < 1$ is a fixed scalar, and

$$\text{dist}_{outside}(\mathbf{a}; \mathbf{p}) = \|\text{Max}(\mathbf{a} - \mathbf{q}_{max}, \mathbf{0}) + \text{Max}(\mathbf{q}_{min} - \mathbf{a}, \mathbf{0})\|_1, \quad (35)$$

$$\text{dist}_{inside}(\mathbf{a}; \mathbf{p}) = \| \text{Cen}(\mathbf{p}) - \text{Min}(\mathbf{q}_{max}, \text{Max}(\mathbf{q}_{min}, \mathbf{a}))\|_1. \quad (36)$$

- **Insideness function** Given query box embeddings $\mathbf{p}_1$ and $\mathbf{p}_2$, the Query2Box insideness function measures if $\mathbf{p}_1$ is inside $\mathbf{p}_2$ by returning the overlap ratio between their intersection and $\mathbf{p}_1$. A higher ratio indicates a greater likelihood that $\mathbf{p}_1$ is inside $\mathbf{p}_2$. In this case, the insideness function is defined as below:

$$\text{Insideness}(\mathbf{p}_1, \mathbf{p}_2) = \frac{\text{BoxVolume}(\text{Intersect}(\mathbf{p}_1, \mathbf{p}_2))}{\text{BoxVolume}(\mathbf{p}_1)} \quad (37)$$

where $\text{BoxVolume}(\mathbf{p})$ measures the volume of the box embedding via a softplus function, such that

$$\text{BoxVolume} = \Pi_i \frac{1}{\beta} log(1 + \exp(\beta \cdot \text{Off}(\mathbf{p})_i)),$$

$$\text{Off}(\mathbf{p})_i \in \text{Off}(\mathbf{p})$$

- **Difference function** Given two boxes, $\mathbf{p}_1$ and $\mathbf{p}_2$, their difference is modeled as

$$\begin{aligned} \text{Diff}(\mathbf{p}_1, \mathbf{p}_2) = \sum_i | \text{Cen}(\mathbf{p}_{1,i}) - \text{Cen}(\mathbf{p}_{2,i}) | + \\ | \text{Off}(\mathbf{p}_{1,i}) - \text{Off}(\mathbf{p}_{2,i}) | \end{aligned} \quad (38)$$

## A.2 BetaE

BetaE represents concepts by the Cartesian product of multiple Beta distributions: $\mathbf{Emb}_C = [(\alpha_1, \beta_1), \ldots, (\alpha_n, \beta_n)]$ where each component is a Beta distribution Beta $(\alpha, \beta)$ controlled with two shape parameters $\alpha$ and $\beta$.

- **Relational Transformation** maps from one Beta embedding S to another Beta embedding S' given the relation type $r$. This is modeled by a transformation neural network for each relation type $r$ using a multi-layer perceptron (MLP):

$$\text{RelT}_r(\mathbf{Emb}_C) = \text{MLP}_r(\mathbf{Emb}_C) \quad (39)$$

- **Intersection Operator** is modeled by taking the weighted product of the PDFs of the input Beta embeddings

$$\text{Intersect}(\mathbf{Emb}_{C_1}, \cdots, \mathbf{Emb}_{C_n}) = \frac{1}{Z} \prod p_{\mathbf{Emb}_C}^{w_1} \cdots p_{\mathbf{Emb}_C}^{w_n} \quad (40)$$

where $Z$ is a normalization constant and $w_1, \cdots, w_n$ are the weights with their sum equal to 1.
- **Complement Operator** is modeled by taking the reciprocal of the shape parameters.

$$\text{Complement}(\mathbf{Emb}_C) = \left[ \left( \frac{1}{\alpha_1}, \frac{1}{\beta_1} \right), \ldots, \left( \frac{1}{\alpha_n}, \frac{1}{\beta_n} \right) \right] \quad (41)$$

- **Distance function** Given an answer entity embedding $\mathbf{a}$ with parameters $\left[ \left( \alpha_1^a, \beta_1^a \right), \ldots, \left( \alpha_n^a, \beta_n^a \right) \right]$, and a query embedding $\mathbf{q}$ with parameters $\left[ \left( \alpha_1^q, \beta_1^q \right), \ldots, \left( \alpha_n^q, \beta_n^q \right) \right]$, we define the distance between this entity $a$ and the query $q$ as the sum of KL divergence between the two Beta embeddings along each dimension:

$$\text{dist}_{beta}(a; q) = \sum_{i=1}^n \text{KL}\left( p_{\mathbf{a,i}}; p_{\mathbf{q,i}} \right) \quad (42)$$

- **Difference function** Given two beta query embeddings, $\mathbf{Emb}_{C_1} = [(\alpha_{11}, \beta_{11}), \ldots, (\alpha_{1n}, \beta_{1n})]$ and

$\text{Emb}_{C_2} = [(\alpha_{21}, \beta_{21}), \ldots, (\alpha_{2n}, \beta_{2n})]$, their difference is modeled as

$$\text{Diff}_{beta}(\text{Emb}_{C_1}, \text{Emb}_{C_2}) = \sum_{i \in \{1, \cdots, n\}} | \alpha_{1i} - \alpha_{2i} | + $$
$$| \beta_{1i} - \beta_{2i} | \quad (43)$$

- **insideness function** Given query beta embeddings $\mathbf{q}_1$ and $\mathbf{q}_2$, the BetaE insideness function meansures if $\mathbf{q}_1$ is inside $\mathbf{q}_2$ by returning the difference between their intersection and $\mathbf{q}_1$. Their intersection is expected to match $\mathbf{q}_1$ if $\mathbf{q}_1$ is fully inside $\mathbf{q}_2$. Thus, the beta insideness function can be defined as

$$\text{Insideness}_{beta} = \text{Diff}_{beta}(\text{Intersect}_{beta}(\mathbf{q}_1, \mathbf{q}_2), \mathbf{q}_1) \quad (44)$$

## A.3 ConE

ConE model concepts by a Cartesian product of sector-cones. Specifically, ConE uses the parameter $\theta_{\text{ax}}^i$ to represent the semantic center, and the parameter $\theta_{\text{ap}}^i$ to determine the boundary of the query. Given a $d$-ary Cartesian product, the embedding of concept is defined as

$$\text{Emb}_C = \text{MultiCone}\left(\boldsymbol{\theta}_{\text{ax}}, \boldsymbol{\theta}_{\text{ap}}\right) \quad (45)$$

where $\boldsymbol{\theta}_{\text{ax}} \in [-\pi, \pi)^d$ are axes and $\boldsymbol{\theta}_{\text{ap}} \in [0, 2\pi]^d$ are apertures.

- **Nominal** is defined as a cone with apertures 0.

$$\text{Nominal}(\text{Emb}_a) = \text{MultiCone}(\boldsymbol{\theta}_{\text{ax}}, 0) \quad (46)$$

- **Relational Transformation** maps a cone embedding to another cone embedding. This is implemented by a relation specific transformation.

$$\text{RelT}(_r\left(\text{MultiCone}(\boldsymbol{\theta}_{\text{ax}}, \boldsymbol{\theta}_{\text{ap}})\right) = g\left(\text{MLP}\left(\left[\boldsymbol{\theta}_{\text{ax}} + \boldsymbol{\theta}_{\text{ax},r}; \boldsymbol{\theta}_{\text{ap}} + \boldsymbol{\theta}_{\text{ap},r}\right]\right)\right) \quad (47)$$

- **Intersection Operator** Suppose that $\text{Emb}_C = \left(\boldsymbol{\theta}_{\text{ax}}, \boldsymbol{\theta}_{\text{ap}}\right)$ and $\text{Emb}_{C_i} = \left(\boldsymbol{\theta}_{i,\text{ax}}, \boldsymbol{\theta}_{i,\text{ap}}\right)$ are cone embeddings for $C$ and $C_i$, respectively. We define the intersection operator as follows:

$$\boldsymbol{\theta}_{\text{ax}} = \text{SemanticAverage}\left(\mathbf{V}_{q_1}^c, \ldots, \mathbf{V}_{q_n}^c\right)$$
$$\boldsymbol{\theta}_{\text{ap}} = \text{CardMin}\left(\mathbf{V}_{q_1}^c, \ldots, \mathbf{V}_{q_n}^c\right) \quad (48)$$

where SemanticAverage $(\cdot)$ and CardMin$(\cdot)$ generates semantic centers and apertures, respectively.

- **Complement Operator** Suppose that $\text{Emb}_C = \text{MultiCone}\left(\boldsymbol{\theta}_{\text{ax}}, \boldsymbol{\theta}_{\text{ap}}\right)$ and $\text{Emb}_{\neg C} = \text{MultiCone}\left(\boldsymbol{\theta}'_{\text{ax}}, \boldsymbol{\theta}'_{\text{ap}}\right)$. We define the complement operator as:

$$[\boldsymbol{\theta}'_{\text{ax}}]_i = \begin{cases} [\boldsymbol{\theta}_{\text{ax}}]_i - \pi, & \text{if } [\boldsymbol{\theta}_{\text{ax}}]_i \geq 0 \\ [\boldsymbol{\theta}_{\text{ax}}]_i + \pi, & \text{if } [\boldsymbol{\theta}_{\text{ax}}]_i < 0 \end{cases}$$
$$\left[\boldsymbol{\theta}'_{\text{ap}}\right]_i = 2\pi - [\boldsymbol{\theta}_{\text{ap}}]_i . \quad (49)$$

- **Distance function.** Suppose that the entity embedding is $\mathbf{v} = (\theta_{\text{ax}}^v, \mathbf{0})$, and the query cone embedding is $\mathbf{V}_q^c =$

$(\boldsymbol{\theta}_{\text{ax}}, \boldsymbol{\theta}_{\text{ap}})$, $\boldsymbol{\theta}_L = \boldsymbol{\theta}_{\text{ax}} - \boldsymbol{\theta}_{\text{ap}}/2$ and $\boldsymbol{\theta}_U = \boldsymbol{\theta}_{\text{ax}} + \boldsymbol{\theta}_{\text{ap}}/2$. The distance between the query and the entity is defined as

$$d_{con}(\mathbf{v}; \mathbf{V}_q^c) = d_o(\mathbf{v}; \mathbf{V}_q^c) + \lambda d_i(\mathbf{v}; \mathbf{V}_q^c). \quad (50)$$

The outside distance and the inside distance are

$$d_o = \left\|\min\left\{\left|\sin\left(\boldsymbol{\theta}_{\text{ax}}^v - \boldsymbol{\theta}_L\right)/2\right|, \left|\sin\left(\boldsymbol{\theta}_{\text{ax}}^v - \boldsymbol{\theta}_U\right)/2\right|\right\}\right\|_1,$$
$$d_i = \left\|\min\left\{\left|\sin\left(\boldsymbol{\theta}_{\text{ax}}^v - \boldsymbol{\theta}_{\text{ax}}\right)/2\right|, \left|\sin\left(\boldsymbol{\theta}_{\text{ap}}\right)/2\right|\right\}\right\|_1,$$

where $\|\cdot\|_1$ is the $L_1$ norm, $\sin(\cdot)$ and $\min(\cdot)$ are element-wise sine and minimization functions.

- **Difference function** Given two cone embeddings, $\text{Emb}_{C1} = \text{MultiCone}\left(\boldsymbol{\theta}_{\text{ax},1}, \boldsymbol{\theta}_{\text{ap},1}\right)$ and $\text{Emb}_{C2} = \text{MultiCone}\left(\boldsymbol{\theta}_{\text{ax},2}, \boldsymbol{\theta}_{\text{ap},2}\right)$, their difference is modeled as

$$\text{Diff}_{cone}(\text{Emb}_{C_1}, \text{Emb}_{C_2}) = \sum_{i \in \{1, \cdots, n\}} | \theta_{\text{ax},1i} - \theta_{\text{ax},2i} | + $$
$$| \theta_{\text{arg},1i} - \theta_{\text{arg},2i} | \quad (51)$$

- **insideness function** Given query cone embeddings $\mathbf{q}_1$ and $\mathbf{q}_2$, the ConE insideness function meansures if $\mathbf{q}_1$ is inside $\mathbf{q}_2$ by returning the difference between their intersection and $\mathbf{q}_1$. Their intersection is expected to match $\mathbf{q}_1$ if $\mathbf{q}_1$ is fully inside $\mathbf{q}_2$. Thus, the ConE insideness function can be defined as

$$\text{Insideness}_{cone} = \text{Diff}_{cone}(\text{Intersect}(\mathbf{q}_1, \mathbf{q}_2), \mathbf{q}_1) \quad (52)$$

## B Interpretation of $\mathcal{ALCOIR}$ Descriptions

Table 5 presents how $\mathcal{ALCOIR}$ concept and role descriptions are interpreted. Given an interpretation $\mathcal{I}$, for each description $X$ the interpretation of the description $X^{\mathcal{I}}$ is recursively defined.

**Table 5: Interpretation of $\mathcal{ALCOIR}$ Descriptions. On the left are concept or role descriptions $X$, and on the right are the interpretations $X^{\mathcal{I}}$.**

| $X$ | $X^{\mathcal{I}}$ |
|---|---|
| $\top$ | $\Delta^{\mathcal{I}}$ |
| $\{a\}$ | $\{a^{\mathcal{I}}\}$ |
| $C \sqcap D$ | $C^{\mathcal{I}} \cap D^{\mathcal{I}}$ |
| $\neg C$ | $\Delta^{\mathcal{I}} \setminus C^{\mathcal{I}}$ |
| $\exists R.C$ | $\{u \mid (u,v) \in R^{\mathcal{I}} \text{ and } v \in C^{\mathcal{I}}\}$ |
| $R^-$ | $\{(u,v) \mid (v,u) \in R^{\mathcal{I}}\}$ |
| $R \circ S$ | $\{(u,v) \mid \text{exists } w, (u,w) \in R^{\mathcal{I}} \text{ and } (w,v) \in S^{\mathcal{I}}\}$ |
| $R \sqcap S$ | $\{(u,v) \mid (u,v) \in R^{\mathcal{I}} \text{ and } (u,v) \in S^{\mathcal{I}}\}$ |
| $R^+$ | $\bigcup_{i \geq 1}(R^{\mathcal{I}})^i$, i.e., $R^+$ is the transitive closure of $R^I$ |

## C Computation Graphs of DAG queries

In this appendix, we define a graph representation for DAG queries.

*Definition C.1 (Computation Graph Role Composition).* The *role composition* of a computation graph $\Gamma = (N, E, \lambda, \tau)$ with a role description $R$, denoted $\Gamma[E]$, is the computation graph $(N', E', \lambda', \tau')$ defined recursively as follows:

(1) If the role description $R$ is a role name $r \in \mathcal{R}$ or the inverse $r^-$ of a role name $r$ then:
  (a) $N' = N \cup \{u\}$ where $u \notin N$;
  (b) $E' = E \cup \{(\tau, u)\}$;
  (c) $\lambda' = \lambda \cup \{u \mapsto \exists R\}$; and
  (d) $\tau' = u$.
(2) $\Gamma[R \circ S] = \Gamma[R][S]$.
(3) $\Gamma[R^{--}] = \Gamma[R]$.
(4) $\Gamma[(R \circ S)^-] = \Gamma[S^- \circ R^-]$.
(5) $\Gamma[(R \sqcap S)^-] = \Gamma[R^- \sqcap S^-]$.
(6) Let $R_1$ and $R_2$ be two role descriptions, $\Gamma[R_1]$ be $(N_1, E_1, \lambda_1, \tau_1)$ and $\Gamma[R_2]$ be $(N_2, E_2, \lambda_2, \tau_2)$. If $R$ is $R_1 \sqcap R_2$ then:
  (a) $N' = N_1 \cup N_2 \cup \{u\}$ where $N_1 \cap N_2 = N$ and $u \notin N_1 \cup N_2$;
  (b) $E' = E_1 \cup E_2 \cup \{(\tau_1, u), (\tau_2, u)\}$;
  (c) $\lambda' = \lambda_1 \cup \lambda_2 \cup \{u \mapsto \sqcap\}$;
  (d) $\tau' = u$.

*Definition C.2 (DAG computation graph).* The *computation graph* of a DAG query $Q$ is the smallest computation graph $\Gamma(C) = (N(Q), E(Q), \lambda(Q), \tau(Q))$ defined recursively as follows:

(1) If $Q$ is $\{a\}$ then:
  (a) $N(Q) = \{u\}$;
  (b) $E(Q) = \emptyset$;
  (c) $\lambda(Q) = \{u \mapsto \{a\}\}$; and
  (d) $\tau(Q) = u$.
(2) If $Q$ is $C \sqcap D$ then:
  (a) $N(Q) = N(C) \cup E(D) \cup \{u\}$ where the sets $N(C), E(D)$, and $\{u\}$ are pairwise disjoint.
  (b) $E(Q) = E(C) \cup E(D) \cup \{(\tau(C), u), (\tau(D), u)\}$
  (c) $\lambda(Q) = \lambda(C) \cup \lambda(D) \cup \{u \mapsto \sqcap\}$.
  (d) $\tau(Q) = u$.
(3) If $Q$ is $\neg C$ then:
  (a) $N(Q) = N(C) \cup \{u\}$ where $N(C) \cap \{u\} = \emptyset$.
  (b) $E(Q) = E(C) \cup \{(\tau(C), u)\}$.
  (c) $\lambda(Q) = \lambda(C) \cup \{u \mapsto \neg\}$.
  (d) $\tau(Q) = u$.
(4) If $Q$ is $\exists R.C$ then $\Gamma(Q) = \Gamma(C)[R]$.

# D  Relaxed tree-form query types

Figure 4 illustrates the query graphs of the new DAG query types and their corresponding relaxed tree-form query types.

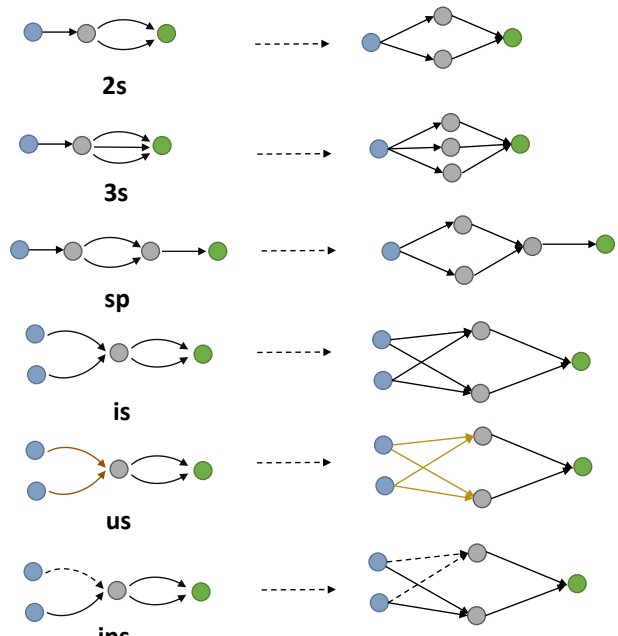

**Figure 4: Transformation from DAG query to relaxed tree-form query**

Each of these queries are expressed as $\mathcal{ALCOIR}$ concepts as follows:

$$2s ::= \exists(r_1 \circ (r_2 \sqcap r_3))^-.\{e_1\}, \tag{53}$$

$$\mathrm{tree}(2s) ::= \exists(r_1 \circ r_2)^-.\{e_1\} \sqcap \tag{54}$$
$$\exists(r_1 \circ r_3)^-.\{e_1\},$$

$$3s ::= \exists(r_1 \circ (r_2 \sqcap r_3 \sqcap r_4))^-.\{e_1\}, \tag{55}$$

$$\mathrm{tree}(3s) ::= \exists(r_1 \circ r_2)^-.\{e_1\} \sqcap \tag{56}$$
$$\exists(r_1 \circ r_3)^-.\{e_1\} \sqcap$$
$$\exists(r_1 \circ r_4)^-.\{e_1\},$$

$$sp ::= \exists(r_1 \circ (r_2 \sqcap r_3) \circ r_4)^-.\{e_1\}, \tag{57}$$

$$\mathrm{tree}(sp) ::= \exists(r_1 \circ r_2 \circ r_4)^-.\{e_1\} \sqcap \tag{58}$$
$$\exists(r_1 \circ r_3 \circ r_4)^-.\{e_1\},$$

$$is ::= \exists(r_3 \sqcap r_4)^-.(\exists r_1\{e_1\} \sqcap \exists r_2\{e_2\}), \tag{59}$$

$$\mathrm{tree}(is) ::= \exists r_3^-.(\exists r_1.\{e_1\} \sqcap \exists r_2\{e_2\}) \sqcap \tag{60}$$
$$\exists r_4^-.(\exists r_1\{e_1\} \sqcap \exists r_2\{e_2\}),$$

$$us ::= \exists(r_3 \sqcap r_4)^-.(\exists r_1\{e_1\} \sqcup \exists r_2\{e_2\}), \tag{61}$$

$$\mathrm{tree}(us) ::= \exists r_3^-.(\exists r_1.\{e_1\} \sqcup \exists r_2\{e_2\}) \sqcap \tag{62}$$
$$\exists r_4^-.(\exists r_1\{e_1\} \sqcup \exists r_2\{e_2\}),$$

$$ins ::= \exists(r_3 \sqcap r_4)^-.(\exists r_1\{e_1\} \sqcap \neg \exists r_2\{e_2\}), \tag{63}$$

$$\mathrm{tree}(ins) ::= \exists r_3^-.(\exists r_1.\{e_1\} \sqcap \neg \exists r_2\{e_2\}) \sqcap \tag{64}$$
$$\exists r_4^-.(\exists r_1\{e_1\} \sqcap \neg \exists r_2\{e_2\}).$$

**Table 6: Number of train/valid/test queries generated for individual DAG query structure in easy and hard modes.**

| Dataset | Train | Valid | Test-Easy | Test-Hard |
|---|---|---|---|---|
| NELL-DAG | 10,000 | 1000 | 1000 | 1500 |
| FB15k-237-DAG | 50,000 | 1000 | 5000 | 4700 |
| FB15k-DAG | 80,000 | 8000 | 8000 | 7500 |

## E  Additional analyses on FB15k-237-DAG-QA and FB15k-DAG-QA datasets

Figures 5 and 6 provide additional analysis on the answer sets of the randomly generated DAG queries from FB15k and FB15k-237.

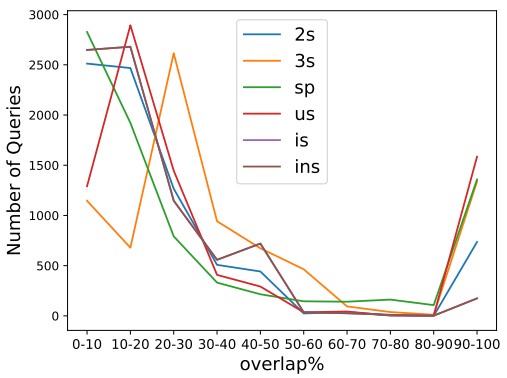

**Figure 5: Proportion of overlap between DAG query answers and the corresponding Tree-Form query answers from FB15k-DAG-QA Easy test dataset.**

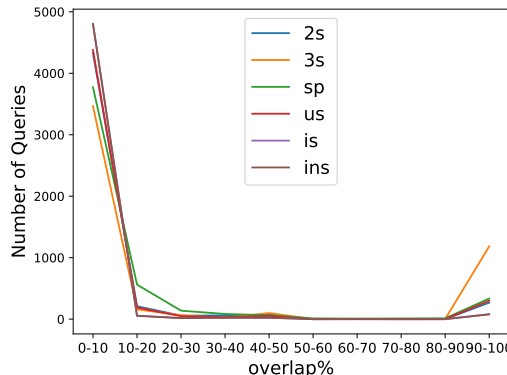

**Figure 6: Proportion of overlap between DAG query answers and the corresponding Tree-Form query answers from FB15k-237-DAG-QA Easy test dataset.**

## F  Further experimental details

### F.1  Hyperparameters and Computational Resource

All of our experiments are implemented in Pytorch [10] framework and run on four Nvidia A100 GPU cards. For hyperparameters search, we performed a grid search of learning rates in $\{5 \times 10^{-5}, 10^{-4}, 5 \times 10^{-4}\}$, the batch size in $\{256, 512, 1024\}$, the negative sample sizes in $\{128, 64\}$, the regularization coefficient $\omega$ in $\{0.02, 0.05, 0.08, 0.1\}$ and the margin $\gamma$ in $\{10, 16, 24, 30, 40, 60, 80\}$. The best hyperparameters are shown in Table 7.

| Dataset | Model | d | b | n | $\gamma$ | l | $\omega$ |
|---|---|---|---|---|---|---|---|
| NELL-DAG | Query2Box (DAGE) | 400 | 512 | 128 | 24 | 1e-4 | - |
|  | BetaE (DAGE) | 400 | 512 | 128 | 60 | 1e-4 | - |
|  | ConE (DAGE) | 800 | 512 | 128 | 20 | 1e-4 | 0.02 |
| FB15k-237-DAG | Query2Box (DAGE) | 400 | 512 | 128 | 16 | 1e-4 | - |
|  | BetaE (DAGE) | 400 | 512 | 128 | 60 | 1e-4 | - |
|  | ConE (DAGE) | 800 | 512 | 128 | 30 | 5e-5 | 0.02 |
| FB15k-DAG | Query2Box (DAGE) | 400 | 512 | 128 | 16 | 1e-4 | - |
|  | BetaE (DAGE) | 400 | 512 | 128 | 60 | 1e-4 | - |
|  | ConE (DAGE) | 800 | 512 | 128 | 40 | 5e-5 | 0.02 |

**Table 7: Hyperparameters found by grid search. d is the embedding dimension, b is the batch size, n is the negative sampling size, $\gamma$ is the margin in loss, l is the learning rate, $\omega$ is the regularization parameter in the distance function.**

### F.2  Further implementation details of DAGE with additional constraints

For the regularization of the restricted conjunction preserving tautology, we encourage the tautology $\exists(r \sqcap s).\{e\} \equiv \exists r.\{e\} \sqcap \exists s.\{e\}$ (see Proposition 3.2) with the following loss:

$$\mathcal{L}_r ::= \mathrm{Diff}(\mathbf{Emb}_{\exists(r\sqcap s).\{e\}}, \mathrm{Intersect}(\mathbf{Emb}_{r.\{e\}}, \mathbf{Emb}_{s.\{e\}})), \quad (65)$$

To enforce the minimization of such loss in our learning objective, we further mine two types of queries from the existing train queries, 2rs and 3rs, that can be expressed as $\mathcal{ALCOIR}$ concepts as follows:

$$2rs ::= \exists((r_1 \sqcap r_2))^-.\{e_1\}, \quad (66)$$

$$\mathrm{tree}(2rs) ::= \exists(r_1)^-.\{e_1\} \sqcap \quad (67)$$
$$\exists(r_2)^-.\{e_1\},$$

$$3rs ::= \exists((r_1 \sqcap r_2 \sqcap r_3)^-.\{e_1\}, \quad (68)$$

$$\mathrm{tree}(3rs) ::= \exists(r_1)^-.\{e_1\} \sqcap \quad (69)$$
$$\exists(r_2)^-.\{e_1\} \sqcap$$
$$\exists(r_3)^-.\{e_1\}.$$

### F.3  Computational costs of DAGE

To evaluate the training speed, for each model with DAGE, we calculated the average running time (RT) per 100 training steps on dataset NELL-DAG. For fair comparison with baseline models, we ran all models with the same number of embedding parameters. Integrating DAGE generally increases the computational cost for existing models. However, models like Query2Box can be enhanced

to outperform baseline models like BetaE, while still maintaining lower computational costs of 22s per 100 steps.

**Table 8: Computational costs of DAGE and the baselines.**

| Model | AVG MRR | RT per 100 steps |
|---|---|---|
| Q2B[15] | 21.23 | 15s |
| Q2B+DAGE | **42.41** | 22s |
| BetaE[3] | 18.32 | 24s |
| BetaE+DAGE | 41.33 | 37s |
| ConE[2] | 26.54 | 18s |
| ConE+DAGE | 40.19 | 50s |

## G Performance of DAGE on Tree-form queries

Table 9 summarizes the performances of query embedding models with DAGE on existing tree-form query answering benchmark datasets, NELL-QA, FB15k-237-QA and FB15k-QA [3]. Firstly, DAGE enhances these models by enabling them to handle DAG queries while preserving their original performance on tree-form queries.

Secondly, DAGE shows significant improvement only on DAG queries, with little effect on tree-form queries, supporting our assumption that DAGE effectively enhances baseline performance for these new query types.

## H Comparison with other query embedding methods

To further assess the effectiveness of DAGE, we compare the baseline models enhanced by DAGE with two prominent query embedding models, CQD [11] and BiQE [4]. Both models are theoretically believed to be capable of handling DAG queries based on their design. Table 10 and 11 summarize the performances of these models on our proposed DAG queries benchmark datasets. These methods enhanced with DAGE consistently outperform CQD and BiQE across all types of DAG queries and datasets. Although some baseline methods perform significantly worse than BiQE and CQD in tree-form query answering tasks (according to the reported results from BiQE and CQD), their integration with DAGE allows them to improve and surpass those models on the new DAG query benchmark datasets. This further supports our argument regarding the effectiveness of DAGE.

| Dataset | Model | 1p | 2p | 3p | 2i | 3i | pi | ip | 2u | up | AVG |
|---|---|---|---|---|---|---|---|---|---|---|---|
| NELL-QA | Q2B | 42.7 | 14.5 | 11.7 | 34.7 | 45.8 | 23.2 | 17.4 | 12.0 | 10.7 | 23.6 |
| | Q2B (DAGE) | 42.09 | 23.39 | 21.28 | 28.64 | 41.09 | 20.0 | 12.30 | 27.51 | 15.86 | 28.3 |
| | BetaE | 53.0 | 13.0 | 11.4 | 37.6 | 47.5 | 24.1 | 14.3 | 12.2 | 8.5 | 24.6 |
| | BetaE (DAGE) | 53.4 | 12.9 | 10.8 | 37.6 | 47.1 | 23.8 | 13.8 | 12.3 | 8.3 | 24.4 |
| | ConE | 53.1 | 16.1 | 13.9 | 40.0 | 50.8 | 26.3 | 17.5 | 15.3 | 11.3 | 27.2 |
| | ConE (DAGE) | 53.2 | 15.7 | 13.7 | 39.9 | 50.7 | 26.0 | 17.0 | 14.8 | 10.9 | 26.8 |
| FB15k-237-QA | Q2B | 41.3 | 9.9 | 7.2 | 31.1 | 45.4 | 21.9 | 13.3 | 11.9 | 8.1 | 21.1 |
| | Q2B (DAGE) | 42.6 | 11.4 | 9.3 | 30.2 | 42.8 | 22.4 | 12.1 | 12.1 | 9.2 | 21.4 |
| | BetaE | 39.0 | 10.9 | 10.0 | 28.8 | 42.5 | 22.4 | 12.6 | 12.4 | 9.7 | 20.9 |
| | BetaE (DAGE) | 38.9 | 10.87 | 9.94 | 29.1 | 42.7 | 22.0 | 11.0 | 12.1 | 9.6 | 20.7 |
| | ConE | 41.8 | 12.8 | 11.0 | 32.6 | 47.3 | 25.5 | 14.0 | 14.5 | 10.8 | 23.4 |
| | ConE (DAGE) | 42.13 | 12.8 | 10.8 | 32.6 | 47.0 | 25.4 | 13.2 | 14.3 | 10.5 | 23.2 |
| FB15k-QA | Q2B | 70.5 | 23.0 | 15.1 | 61.2 | 71.8 | 41.8 | 28.7 | 37.7 | 19.0 | 40.1 |
| | Q2B (DAGE) | 67.9 | 24.5 | 21.3 | 53.4 | 64.82 | 40.3 | 23.5 | 35.2 | 21.7 | 39.2 |
| | BetaE | 65.1 | 25.7 | 24.7 | 55.8 | 66.5 | 43.9 | 28.1 | 40.1 | 25.2 | 41.6 |
| | BetaE (DAGE) | 64.5 | 24.6 | 23.6 | 55.6 | 66.5 | 42.8 | 22.5 | 40.2 | 23.8 | 40.5 |
| | ConE | 73.3 | 33.8 | 29.2 | 64.4 | 73.7 | 50.9 | 35.7 | 55.7 | 31.4 | 49.8 |
| | ConE (DAGE) | 74.3 | 31.9 | 27.4 | 63.9 | 73.5 | 50.1 | 29.8 | 53.6 | 29.4 | 48.2 |

**Table 9: MRR performance of the retrained baseline models with DAGE method on tree-form query benchmark datasets**

| Dataset | Model | 2s | 3s | sp | is | us | Avg$_{nn}$ | ins | Avg |
|---|---|---|---|---|---|---|---|---|---|
| NELL-DAG | Query2Box (DAGE) | **37.61** | 49.42 | **41.71** | **40.57** | **42.75** | **42.41** | - | - |
| | BetaE (DAGE) | 36.87 | 57.14 | 34.95 | 39.90 | 37.80 | 41.33 | **34.68** | **40.22** |
| | ConE (DAGE) | 33.50 | **57.35** | 38.43 | 37.93 | 33.74 | 40.19 | 33.94 | 39.15 |
| | CQD-Beam | 22.60 | 44.84 | 17.51 | 24.88 | 1.60 | 22.29 | - | - |
| | BiQE | 20.74 | 45.38 | 20.76 | 28.37 | - | 28.81 | - | - |
| FB15k-237-DAG | Query2Box (DAGE) | **7.41** | **12.64** | 10.07 | **7.32** | **5.03** | 8.49 | - | - |
| | BetaE (DAGE) | 6.27 | 12.11 | 9.64 | 6.66 | 4.09 | 7.75 | **6.58** | 7.56 |
| | ConE (DAGE) | 6.87 | 11.66 | **12.36** | 6.90 | 4.80 | **9.54** | 6.08 | **8.12** |
| | CQD-Beam | 4.31 | 8.65 | 5.56 | 3.71 | 0.13 | 4.47 | - | - |
| | BiQE | 2.11 | 2.74 | 4.08 | 5.73 | - | 3.67 | - | - |
| FB15k-DAG | Query2Box (DAGE) | 37.74 | 42.93 | 24.30 | 29.37 | 25.97 | 31.46 | - | - |
| | BetaE (DAGE) | 32.65 | 46.17 | 32.48 | 28.15 | 28.10 | 33.50 | 25.39 | 32.15 |
| | ConE (DAGE) | **41.67** | **56.70** | **33.36** | **36.54** | **32.36** | **40.12** | 30.86 | **38.58** |
| | CQD-Beam | 22.21 | 36.77 | 23.44 | 15.18 | 1.64 | 19.85 | - | - |
| | BiQE | 26.31 | 35.12. | 20.37 | 20.08 | - | 22.25 | - | - |

**Table 10: The table presents the MRR performance of baseline models integrated with DAGE on easy benchmark datasets, in comparison with CQD[11] and BiQE[4].**

| Dataset | Model | 2s | 3s | sp | is | us | Avg$_{nn}$ | ins | Avg |
|---|---|---|---|---|---|---|---|---|---|
| NELL-DAG | Query2Box (DAGE) | 25.38 | 20.13 | 21.25 | 24.85 | 29.24 | 24.17 | - | - |
| | BetaE (DAGE) | 27.68 | 32.25 | 16.36 | 26.14 | 29.19 | 26.32 | 33.64 | 27.54 |
| | ConE (DAGE) | **30.71** | **38.41** | **24.76** | **28.44** | **31.06** | **30.67** | **34.07** | **31.24** |
| | CQD-Beam | 12.25 | 21.73 | 10.78 | 12.73 | 2.17 | 11.93 | - | - |
| | BiQE | 13.57 | 19.46 | 12.03 | 15.38 | - | 15.11 | - | - |
| FB15k-237-DAG | Query2Box (DAGE) | 4.81 | **2.81** | 7.87 | **5.26** | **4.38** | **6.95** | - | - |
| | BetaE (DAGE) | **4.89** | 1.66 | 8.28 | 4.75 | 3.50 | 4.61 | **6.06** | 4.85 |
| | ConE (DAGE) | 4.78 | 2.09 | **9.72** | 4.84 | 4.16 | 5.12 | 5.25 | **5.14** |
| | CQD-Beam | 2.74 | 1.63 | 4.63 | 2.38 | 0.10 | 2.29 | - | - |
| | BiQE | 3.13 | 2.01 | 3.37 | 1.89 | - | 2.60 | - | - |
| FB15k-DAG | Query2Box (DAGE) | 33.78 | 39.67 | 19.61 | 26.91 | 24.76 | 28.95 | - | - |
| | BetaE (DAGE) | 30.57 | 44.30 | 29.35 | 25.72 | 26.63 | 31.31 | 25.18 | 30.29 |
| | ConE (DAGE) | **40.14** | **57.06** | 29.23 | **34.63** | **31.45** | **38.50** | **30.74** | **37.21** |
| | CQD-Beam | 19.90 | 33.88 | 22.43 | 12.66 | 1.60 | 18.09 | - | - |
| | BiQE | 17.36 | 30.37 | 25.38 | 15.89 | - | 22.25 | - | - |

Table 11: The table presents the MRR performance of baseline models integrated with DAGE on hard benchmark datasets, in comparison with CQD[11] and BiQE[4].

