# OpenReview forum: "DAGE: DAG Query Answering via Relational Combinator with Logical Constraints"
_ACM.org/TheWebConf/2025/Conference — WWW 2025 Poster_

### Official Review · Reviewer_J3aj · 2024-11-29

**Novelty:** 3
**Technical Quality:** 4

**Review:**

This paper proposes an interesting scenario in the field of knowledge graph complex query answering: complex query reasoning based on DAG structure. The advantages of this paper are as follows:
1. Opened up a new KG complex query answering scenario, which has the potential to help broaden the research field.
2. The internal mechanism of the proposed method was elucidated through specific scenario explanations and solid theoretical proofs.
3. Experiments have verified the effectiveness of the proposed method on DAG format queries.

However, there are still several weaknesses:
1. The motivation analysis of the paper is unclear. The introduction section seems to be introducing the differences between tree-format and DAG-format queries, without clearly stating the advantages of using the DAG-format query. For example, what are the limitations of vanilla methods in strictly constraining intermediate variables? The advantage of relaxing variable constraints in this article's method is the same.
2. Lack of comparison with some query types in public benchmarks.
3. Lack of the latest relevant work and comparative methods.

**Questions:**

1. Would the author mind providing a more focused motivation analysis?

2. As shown in Page 6, Line 688, this paper used BetaE-version datasets as benchmarks and construct new query types (ins) including the negation operation. However, Table 9 only provides the results of EPFO queries. Do authors test the model on the negative queries, such as 2in, 3in, inp, and ipn, in BetaE-version datasets?

3. This paper overlook some important baselines in recent years, such as GNN-QE [1], LMPNN [2], and QTO [3]. They are powerful methods that combine neural linking prediction and fuzzy logic reasoning, which are different with the straightforward query embedding methods. I think the author also needs to validate the effectiveness of DAGE on such methods to demonstrate the universality of the proposed approach.

[1] Zhu, Z., Galkin, M., Zhang, Z., et al. Neural-symbolic models for logical queries on knowledge graphs. ICML 22

[2] Wang, Z., Song, Y., Wong, G. Y., et al. Logical message-passing networks with one-hop inference on atomic formulas. ICLR 23

[3] Bai Y, Lv X, Li J, et al. Answering complex logical queries on knowledge graphs via query computation tree optimization. ICML 23

4. The results in Table 9 demonstrate the negative enhancement effect of the DAGE model on tree format query responses, and I am curious about the reasons behind this.

**Reviewer Confidence:**

3: The reviewer is confident but not certain that the evaluation is correct

**Scope:**

4: The work is relevant to the Web and to the track, and is of broad interest to the community

---

### Official Review · Reviewer_vbk9 · 2024-11-29

**Novelty:** 5
**Technical Quality:** 5

**Review:**

### Pros

- Well-written paper with sufficient background and definitions to present significant difference in tree-form queries based on SROI logic and Directed Acyclic Graph DAG queries based on ALCOIR description logic.
- Novel problem formulation of DAG queries over knowledge graphs.
- New 3 benchmarking datasets (FB15k-DAG, FB15k-237-DAG, NELL-DAG) with 6 types of DAG queries, including at least first-order logical operators such as conjunction, disjunction, there exists in relation projection, and negation.
- DAGE with relation combinator based on DeepSets facilitates to merge multiple paths between two nodes to a single path, and DAGE is a plug-and-play module for existing models (Query2Box, BetaE, ConE).
- Significant improvement on model performance in DAG queries when adding DAGE over the existing models.

### Cons

- The proposed model relies on the neural approach rather than the symbolic methodologies, to handle the conjunction of multiple relational paths in DAG queries, which might limit the interpretation of intermediate entities or quantified variables.
- Minor point: In Table 3, the performance improvement using DAGE of answering original tree-form queries (for non-negation structures) observe in a specific model such as box embeddings, but not in other models such as beta embeddings and cone embeddings.

**Questions:**

- When implementing a model with and without DAGE, is the key difference between a baseline without DAGE and with DAGE in whether using the relational combinator, named in the paper, to merge multiple projection paths? Is there any other differences between these approaches?
- Are there any comments on the generation process of a specific DAG query that is based on the original query in tree-form? For example, 2s (3s) query is based on 2p (3p), etc.
- In the new datasets, do we have two versions for each DAG query structure? Taking the (is) structure for example, a version looks like an example in Eq. 13 (called DAG query), and a relaxed version looks like an example in Eq. 14 (called tree-form query)
- Why do we need to evaluate model performance using DAGE technique, aiming to handle DAG structures, on the original dataset (e.g NELL-QA in Table 3) which is not supposed to have DAG structures? Any motivations here?
- L119-L120: Does DAG denote directed graph query as presented, or is there a typo such as directed **acyclic** graph query?
- L514, Eq. 28: Does this equation compute the similarity of embeddings between a DAG query (an example in Eq. 13) with its simpler version (an example in Eq. 12)?
- L546, Eq. 29: Does this equation compute the difference of embeddings between a DAG query (an example in Eq. 13) with its relaxed version (an example in Eq. 14)?

**Reviewer Confidence:**

4: The reviewer is certain that the evaluation is correct and very familiar with the relevant literature

**Scope:**

3: The work is somewhat relevant to the Web and to the track, and is of narrow interest to a sub-community

---

### Official Review · Reviewer_oSU5 · 2024-12-01

**Novelty:** 5
**Technical Quality:** 6

**Review:**

Authors focuses on improving query embeddings for query answering in the presence of multiple conjunctive paths between two nodes. To do so, the focus in treating the queries as a Directed Graph Queries (DAG), instead of a pure tree-shape. Thus, authors propose a new logic extension that "groups" such conjunctions, and a new relational combinator to extend current embeddings. Finally, they evaluate the approach by extending current well-known benchmarking queries, showing that authors approach (i) improves on the new DAG queries w.r.t the state of the art, and (ii) doesn´t worsen the tree-shape queries

I am not an expert in description logic, but the paper is well written, seems well grounded and technically sound. The results show a reasonable improvement with respect to the state of the art.

Perhaps I have a doubt on what´s the impact of the proposal. As mentioned in the questions section, I wonder whether the scenario covered by authors is realistic and frequent in practice.

**Questions:**

- Can you please clarify the meaning of the acronyms DAG and DAGE? In page 2 (line 119) authors name DAG as Direct Graph Queries, but I wonder if it is related to the concept of Directed Acyclic Graph. No references to DAGE unless I missed it

- Can you please clarify what´s the proportion of DAG queries in a realistic scenario? I was quite surprised that current state of the art and benchmarks disregard these queries.

**Reviewer Confidence:**

2: The reviewer is willing to defend the evaluation, but it is likely that the reviewer did not understand parts of the paper

**Scope:**

3: The work is somewhat relevant to the Web and to the track, and is of narrow interest to a sub-community

---

### Official Review · Reviewer_xZ8k · 2024-12-02

**Novelty:** 5
**Technical Quality:** 7

**Review:**

This paper motivates the need to extend complex query answering approaches to DAG queries. DAG queries differ from tree-form queries allowing for multiple paths between quantified and target variables, and requiring conjunction in role description. The authors argue that DAG queries can’t be handled by existing query embedding methods and propose an extension based on a relational combination operator. This relational combinator is implemented as a weighted sum of role embeddings that were modified through a neural network. Given a set of relations it produces a relation embedding of the same size as the original embeddings. To train this new architecture the authors introduce two new losses that are effectively regularisation terms that encourage tautologies that correspond to the propositions defined for the relational combination operator. To conduct an experimental evaluation of the new approach, the authors generate new synthetic datasets for six different query structures with multiple paths. The experimental results consistently show an improved performance on DAG queries across all query embedding methods. The authors also provide the source code for reproducibility of their experiments.

The paper requires proofreading. Typos:

l.154	Instead, can apply them  -> we can apply them

l.156	this workaround solution produce  -> produces

l.245	a ALCOIR knowledge base -> an

l.260	this is not hold -> does not

l.356	hols -> holds

l.402	DAG-E -> DAGE

l.469	but not not

l.553	regulation terms -> regularisation

l.915	ALCOIR)

l.920 	implementDAGE -> implement DAGE

**Questions:**

1. Based on the example used in the Introduction, the DAG queries constitute a specific subset of the tree-form queries, where x1=x2. Therefore, the baseline approaches with relaxed tree-form query types, as described in Appendix D, will always have a larger answer set both at train time, as positive and negative answer samples, and test time. Are the baselines and DAGE evaluated and trained on the same datasets with the same answer sets and the same number of positive/negative answer samples?

2. What is referred to as DAGE? Is it the approach discussed in Section 4, i.e., the extension to the base query embedding approach with the implementation of the relational combinator as a Deepset of relation embeddings and two new regularisation terms introduced in 4.3? Or DAGE results reported in Tables 1-3 were implemented without regularisation? Making this more explicit in the heading of the section or in its first paragraph would make it more clear.

3. Is it essential to retrain embeddings in DAGE? Will the performance drop if the embedding layers remain frozen (i.e., same embeddings as in baselines) and only the relational combinator’s weights were trained (i.e, only the MLP)?

4. What are the "Avgnn" and "ins" columns in the result tables?

**Ethics Review Flag:**

Yes

**Reviewer Confidence:**

2: The reviewer is willing to defend the evaluation, but it is likely that the reviewer did not understand parts of the paper

**Scope:**

4: The work is relevant to the Web and to the track, and is of broad interest to the community

---

### Official Review · Reviewer_R6bD · 2024-12-03

**Novelty:** 4
**Technical Quality:** 4

**Review:**

This paper introduces a query embedding method called DAGE by generalizing tree-form queries to more relaxed Directed Acyclic Graph (DAG) queries over KGs. This allows for the handling of a more expressive set of queries that correspond to ALCOIR description logic instead of SROI− description logic. DAGE uses a relational combinator to handle the intersection of relations. The authors propose a benchmark for evaluating DAG query embeddings and demonstrate that DAGE improves upon baseline models like Query2Box and BetaE.

Pros:
- The paper proposes a novel method to support DAG queries and introduces a relational combinator grounded on logical theory.
- The authors create datasets specifically for DAG queries that help evaluate complex reasoning over knowledge graphs.
- The paper provides a comprehensive evaluation guided by three research questions as well as an ablation study. The experiments demonstrate gains over the baseline approaches such as Query2Box, BetaE, and ConE.

Cons:
- The construction of test sets with DAG queries should be further explained. It will also be useful to understand the practical impact of supporting DAG queries in real-world KGQA. How often do these queries occur in common KBQA datasets?
- The paper lacks a discussion of the computational overhead introduced by the relational combinator, especially for large-scale knowledge graphs.
- The paper does not mention the limitations of the current approach and does not provide an error analysis of the results to understand the limitations of this approach and possible next steps.

**Questions:**

- What is the computational cost of training/inference in DAGE compared to the baseline methods, particularly for large datasets and have you analyzed the scalability aspect?
- What are your thoughts on extending relational combinator to support additional logical operators?

**Reviewer Confidence:**

2: The reviewer is willing to defend the evaluation, but it is likely that the reviewer did not understand parts of the paper

**Scope:**

3: The work is somewhat relevant to the Web and to the track, and is of narrow interest to a sub-community